# Autonomous discovery of optically active chiral inorganic perovskite nanocrystals through an intelligent cloud lab

Jiagen Li[1], Junzi Li[2], Rulin Liu[1], Yuxiao Tu[1], Yiwen Li[3], Jiaji Cheng ⬚ [3✉], Tingchao He[2✉] & Xi Zhu[1✉]

We constructed an intelligent cloud lab that integrates lab automation with cloud servers and artificial intelligence (AI) to detect chirality in perovskites. Driven by the materials acceleration operating system in cloud (MAOSIC) platform, on-demand experimental design by remote users was enabled in this cloud lab. By employing artificial intelligence of things (AIoT) technology, synthesis, characterization, and parameter optimization can be autonomously achieved. Through the remote collaboration of researchers, optically active inorganic perovskite nanocrystals (IPNCs) were first synthesized with temperature-dependent circular dichroism (CD) and inversion control. The inter-structure (structural patterns) and intra-structure (screw dislocations) dual-pattern-induced mechanisms detected by MAOSIC were comprehensively investigated, and offline theoretical analysis revealed the thermodynamic mechanism inside the materials. This self-driving cloud lab enables efficient and reliable collaborations across the world, reduces the setup costs of in-house facilities, combines offline theoretic analysis, and is practical for accelerating the speed of material discovery.

---

[1] Shenzhen Institute of Artificial Intelligence and Robotics for Society (AIRS), The Chinese University of Hong Kong, Shenzhen (CUHK-Shenzhen), Shenzhen, Guangdong 518172, P.R. China. [2] College of Physics and Optoelectronic Engineering, Shenzhen University, Shenzhen, Guangdong 518060, P. R. China. [3] School of Materials Science and Engineering, Hubei University, Wuhan, Hubei 430062, P. R. China. ✉email: sosjune@163.com; tche@szu.edu.cn; zhuxi@cuhk.edu.cn

In past centuries, scientists have made substantial material discoveries in terms of new element extractions, the development of semiconductors, new drugs to target diseases, renewable energy, and many fields with modern global applications. In this process, large costs were incurred, including economic investment, excessive exploitation, environmental pollution, and even sacrifices made by scientists. However, the rapidly increasing demand for high-performance materials in various industries has not been satisfied. To address this challenge, advanced methodologies for accelerating the discovery of new materials with novel properties are urgently needed. Recently, the fusion of materials science and modern digital technologies, including robotics, cloud computing, and artificial intelligence (AI), has increasingly attracted researcher attention. Software-controlled chemical equipment is widely utilized in laboratories as powerful tools for the automated synthesis of target molecules[1,2], polymers[3], quantum dots (QDs)[4,5], and 2-D materials[6]. These high-throughput automatic tools, which combined with in situ characterization methods, have significantly enhanced the efficiency and reproducibility of material synthesis. Beyond automation, optimization algorithms for closed-loop autonomous discoveries have been widely investigated[7-9]. Autonomous systems that interconnect chemical machines through real-time internet of things (IoT) technologies[10,11], combining theoretical models with AI algorithms[12], liberate researchers from trivial experimental tasks and, more importantly, accelerate the discovery of new materials. Most recently, in order, researchers from Cambridge[13], Glasgow[14], Massachusetts[15,16] and Toronto[17] have reported their intelligent synthetic platforms coupled with AI algorithms, which have proven to be efficient in some high-dimensional optimization tasks with correlated parameters. However, AI-based discoveries of new materials or novel material properties have not been reported; therefore, the expectation of independent material discoveries remains challenging to fulfill.

Recently, chiral perovskites have attracted increasing research attention due to their novel optoelectronic properties since they were first synthesized in 2003[18,19]. Chiral organic-inorganic hybrid perovskites, which possess chirality due to the incorporated chiral organic molecules, were theoretically predicted[20] and synthesized[21-25] with significant optical activity. On the other hand, without electronic interactions with chiral molecules, the design of optically chiral perovskite nanocrystals (NCs) relies solely on symmetry breaking through chiral defects or chiral assembled structures of the inorganic perovskite nanocrystals (IPNCs) themselves. For $CsPbBr_3$ perovskite structures, the high-temperature cubic phase and low-temperature tetragonal and orthorhombic phases are all achiral. Thus, IPNCs with intrinsic chiral properties are seldom reported because of the difficulty of precise control over the enantiomeric synthesis of chiral NCs. Moreover, efficient methods for characterizations and incisive simulations or theoretical support are still lacking.

Herein, we report an intelligent system that successfully discovered optically active $CsPbBr_3$ IPNCs via an autonomous approach. Automation platforms based on flow chemistry, collaborative robots, and in situ characterization techniques are well integrated into the cloud lab. Driven by the customized materials acceleration operating system in cloud (MAOSIC) platform, this cloud lab has provided an authentic synthetic platform for researchers across the world. In this work, by adjusting the 2-dimensional parameter space $[T, C]$ (reaction temperature, precursor concentrations) under the strategies of the stable noisy optimization by branch and fit (SNOBFIT)[26]-based reinforcement learning (RL) algorithm, MAOSIC achieved significant circular dichroism (CD) signal from a $CsPbBr_3$ nanocrystal solution within 250 experimental loops. By combining the direct and indirect evidence from various aspects, the chiral assembled structures and screw dislocation were inferred as the origin of chirality in synthesized rod-like $CsPbBr_3$ nanoplates. Furthermore, we applied femtosecond laser irradiation and found an inverted optical rotation direction of synthesized chiral IPNC samples, indicating a new method for chirality adjustment in inorganic nanostructures. The quantum mechanical theory explained this chirality inversion phenomenon. We expect that this work can provide a vivid demonstration to stimulate materials scientists for generating more ideas for discovering new materials with novel properties by integrating physical theories and AI.

## Results

**Cloud lab architecture**. This lab was constructed according to the framework of the industrial artificial intelligence of things (AIoT) system, as shown in Fig. 1. Robotic automation, cloud servers, sensing devices, and communication tools are indispensable, and the central platform—MAOSIC (upgraded from our previous system MAOS[27])—managed all issues. To complete the complex experimental tasks, five functional modules were integrated within MAOSIC, including (a) a human-machine interaction, (b) a hardware control interface, (c) an analysis module, (d) optimization modules and (e) a cloud control interface (see Supplementary Method 1 for the detailed architectures). The human-machine interaction module provided a web-based user interface (UI) for researchers to check the status, send instructions, and obtain data from experiments through the communication between the cloud and local server. A wireless or 5G network was applied for high-rate data transmission (the communication test is shown in Supplementary Method 2). An encrypted tunnel is built between local equipment and the cloud server to adopt transport layer security (TLS) to encrypt all data transmission. The cloud server also provides a distributed denial of service (DDoS) prevention, which ensures the availability of our system. All users need to use the key built socket shell (SSH) tunnel to connect with the cloud server, with secure access to ensure data security, process safety, and collaboration efficiency. For data storage security, in both the cloud server and local equipment, a Linux unified key was set up to encrypt all files. Driven by the hardware control interface, all chemical machines, collaborative robots, and characterization techniques are managed in efficient teamwork (details of the hardware information are provided in the Methods section and Supplementary Method 3). The hardware control interface in MAOSIC includes both high-level and low-level instructions based on the JSON-RPC2.0 protocol. The high-level instruction is a formula made up of instructional symbols and operators (see the Methods section and Supplementary Method 1). MAOSIC has its own language system for all experimental operations, with the compile process (see the Methods section and Supplementary Method 4). MAOSIC stores long-term data through SQLAlchemy[28], which supports a database management system (DBMS), including MySQL, Postgres, Oracle, and SQLite. MySQL on the cloud server is used to store the data. In this work, as shown in Fig. 1b, a microfluidic reactor with three input channels (syringe pumps) was utilized for IPNC synthesis based on the hot-injection method. The input parameter set $[T$ (°C), $C$ (mmol ml$^{-1}$)] was designed for reaction condition adjustment. The online absorption spectrum was collected in situ during the synthesis process. A collaborative robot and rotation sampler were used to prepare solution samples in quartz cuvettes, where were then transferred to a circular dichroism spectrometer for chirality determination. Detailed synthesis procedures are described in the Methods section. The high-throughput results were preprocessed through the analysis module. For instance, the multipeak separation, peak wavelength,

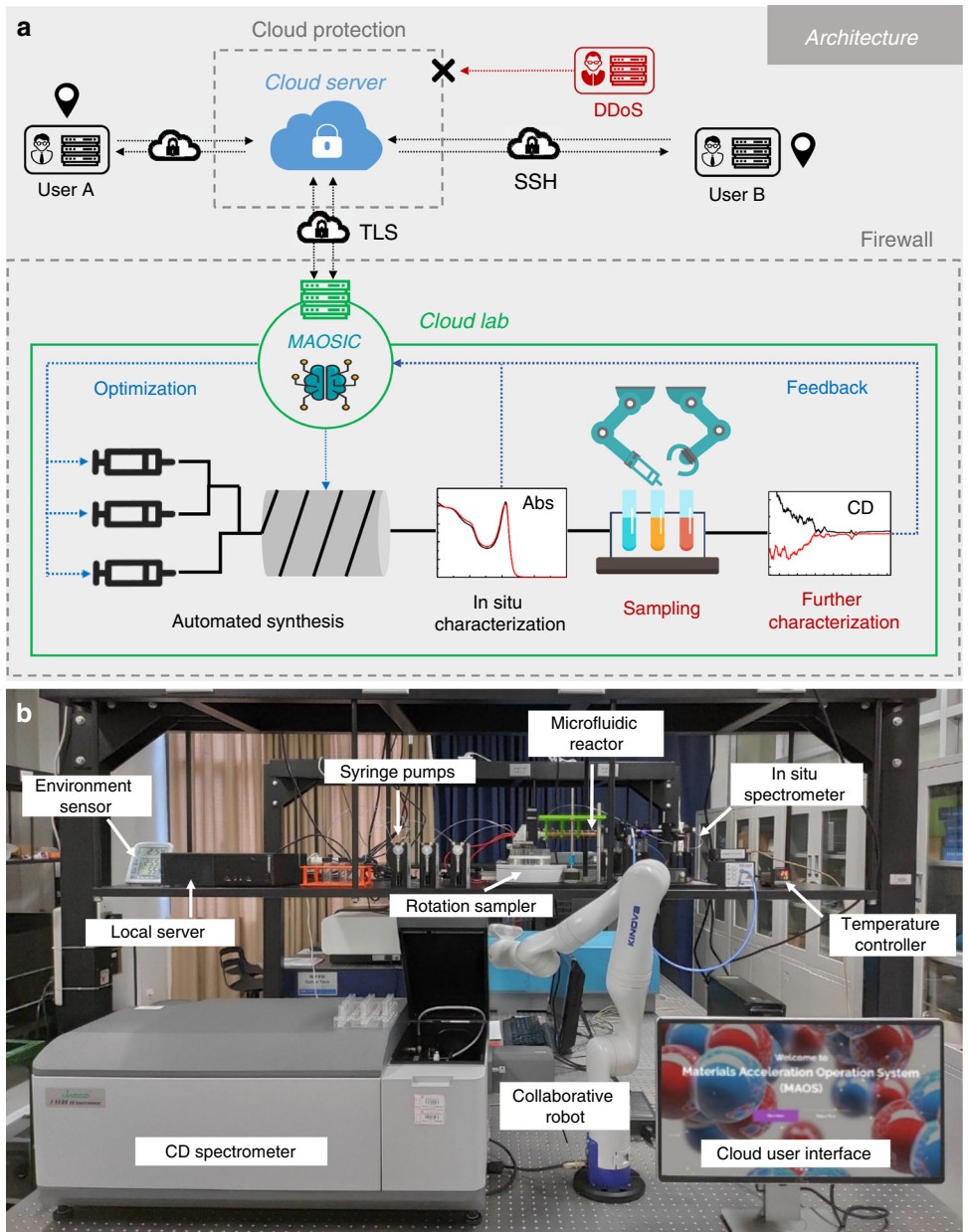

**Fig. 1 Workflow and photograph of the cloud lab for discovering new optically active IPNCs. a** MAOSIC allows remote users to interact with the **b** integrated equipment in the lab through the cloud server. Encrypted communication and firewalls were utilized to maintain data security and transfer stability. Automatic reactions took place in microfluidic reactors and under in situ monitoring through the absorption spectrum. Collaborative robots handled the automatic sampling task for CD measurement. All characterization data were sent back to the analysis and optimization module in MAOSIC for autonomous optimization.

and intensity were determined through Gaussian/Lorenz fitting from the original spectrum. The processed data were then sent to the optimization module to generate synthesis planning for the experiment in the next loop. The RL algorithms and instruction compile process in MAOSIC are presented in the Methods section and Supplementary Method 4.

**Results of autonomous discovery**. Screw dislocation is essential for chirality in various inorganic nanocrystals[29,30]. Therefore, we chose $[T, C]$ as a parameter set to explore the optimal experimental conditions, which has significant effects on screw-dislocation-driven anisotropic crystal growth[31,32]. The sampling points (selected within 250 loops) during the search and optimization processes are shown in Fig. 2a. In short,

SNOBFIT algorithms combine the concepts of random search in the global region and the gradient descent method in the local region[26], making it popular for chemical reaction searches and optimization[33–35]. The 2-D parameter plane allows exploration of synthetic conditions within the preset boundaries ($T \in [90\,°C, 180\,°C]$, $C \in [0.2\,mmol/ml, 1.2\,mmol/ml]$). Guided by algorithms, the light yellow points were sparsely distributed on the 2-D map to divide the parameter space into similar boxes, which indicates the high randomness of sampling at the initial state of the reaction search. The intensity of the CD signal near the first exciton peak was set to evaluate the loss function ($L$) in the formula $L = -|CD|$. The minimum step for adjusting the parameter was set as $[dT = 0.5\,°C, dC = 0.01\,mmol\,ml^{-1}]$ to balance the synthesis accuracy and search

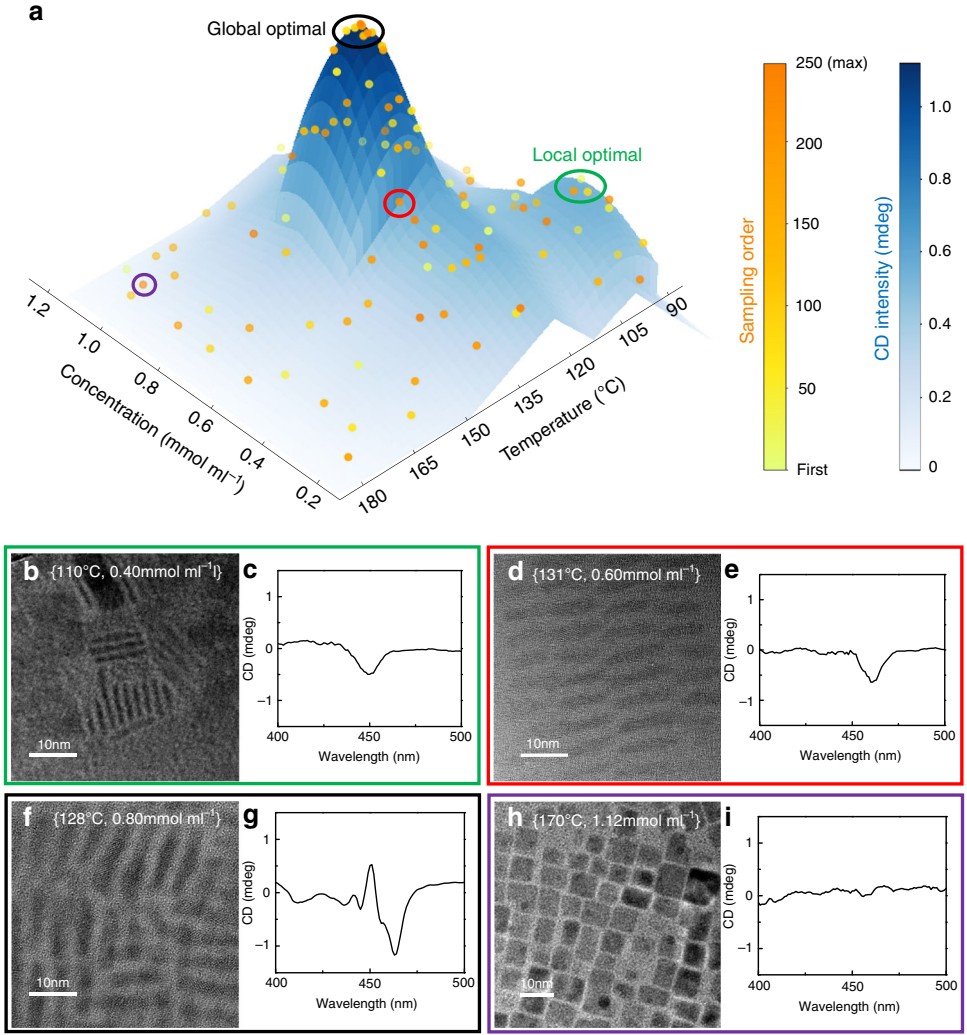

**Fig. 2 The results of autonomously discovered optically active IPNCs driven by MAOSIC. a** SNOBFIT codes guided the chemical machines to search the two-dimensional parameter plane for higher CD signals. **b–i** TEM images and CD spectra of chiral IPNC samples under various synthesis conditions.

efficiency. With ~50 trials, a local optimal region (in the green circle) appeared at a temperature of approximately $[110\,^{\circ}\text{C}, 0.4\,\text{mmol ml}^{-1}]$, with a CD intensity beyond 0.4 mdeg (Fig. 2b, c). The synthesis of IPNCs at these low temperatures could result in significant 1-D growth[36] of nanocrystals with a large aspect ratio. The anisotropic structure provides an extension path for chiral patterns along the Z-axis to enhance the optical activity. However, the poor crystallization and dispersion state limited the sample quality and CD strength. In further trials, another region (in the red circle) with a higher CD intensity appeared, as shown in Fig. 2d, e. Then, a precise gradient descent was applied near the points on the slope, and it achieved a CD strength of 1.0 mdeg (Fig. 2f, g). Up to this point, the search and optimization process required ~120 loops. At the same time, the global random search kept exploring the 2-D parameter space (sparsely distributed dark orange points on the map) and finally reached the preset maximum order (250 loops). The global optimized synthesis condition (in the black circle) was the region of ~120 loops under $[130\,^{\circ}\text{C}, 0.8\,\text{mmol ml}^{-1}]$. The samples in this region have a better crystal quality than those in other regions, as well as an enhanced anisotropic chiral intensity. In the higher temperature region ($T > 160\,^{\circ}\text{C}$, in the purple circle), the QDs appeared in the cubic phase, as shown in Fig. 2h, i. Due to the high symmetry in the cubic phase, there were fewer

chiral assemblies as well as a decreased CD signal intensity compared with those in other phases. According to our computational model (Supplementary Method 5), screw dislocations are more likely to occur in thinner perovskites, which agrees well with these experimental data. For another parameter, namely, the precursor concentration C, we analyzed samples synthesized at ~130 °C and found that the CD intensity near 450 nm (first exciton absorption peak) is proportional to the concentration up to $1.0\,\text{mmol ml}^{-1}$. The high concentration provides more opportunities for overlapping and interaction between nanocrystals, indicating more possibilities of chiral defects than at a low concentration. Since the absorption approaches the limit for the CD spectrometer, we did not investigate concentrations higher than $1.2\,\text{mmol ml}^{-1}$.

**Identify the origins of chirality.** Next, we investigated samples synthesized under local optimal conditions $[110\,^{\circ}\text{C}, 0.4\,\text{mmol ml}^{-1}]$. The X-ray diffraction (XRD) patterns in Fig. 3a demonstrate a significant component of the cubic $CsPbBr_3$ perovskite phase (JCPDS card No. 54-0752). Spindle-shaped chiral assemblies were observed under TEM (Fig. 3b). Due to the poor crystal quality, no clear evidence of screw dislocation can be captured from the blurry lattice strip. The global optimal sample (synthesized at $[130\,^{\circ}\text{C}, 0.8\,\text{mmol ml}^{-1}]$) was characterized by HRTEM (Fig. 3d), and the size

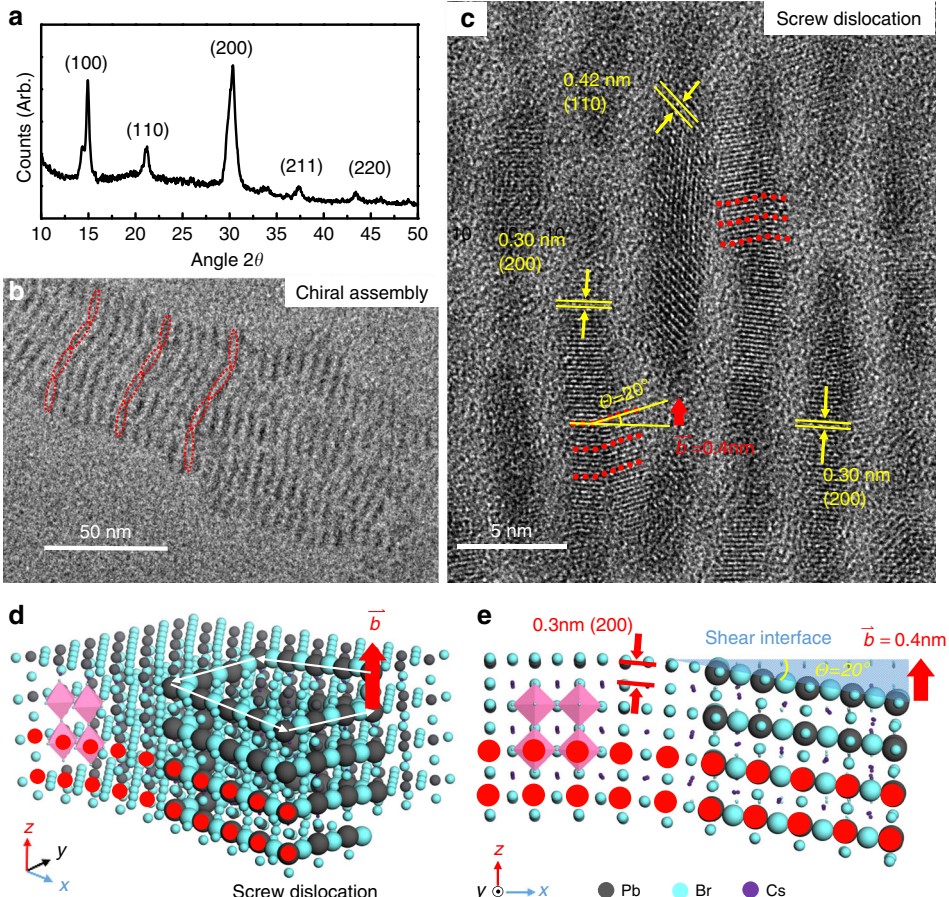

**Fig. 3 Structure identification of the synthesized chiral CsPbBr₃ nanocrystal sample. a** The cubic phase of crystalline CsPbBr₃ was determined from the X-ray diffraction (XRD) pattern. **b** The chiral assembly of rod-like CsPbBr₃ nanoplates (local optimal sample) was found by transmission electron microscopy (TEM) images. **c** Evidence of screw dislocation measured by high-resolution TEM (HRTEM) in the global optimal sample. The (110) and (200) phases were measured to confirm the cubic crystal structure. The curved lattices along the (200) phase are marked with red points corresponding to the TEM image, which indicates the possible screw dislocation in the nanocrystal, the schematic in (**d**) and (**e**). The screw line is highlighted with amplified Pb–Br atoms, perpendicular to the Burgers vector ($\vec{b}$ = 0.4 nm). The shear interface with a bent angle ($\theta$ = 20°) is perpendicular to the (200) phase (drawn in blue).

of these nanoplates was ~4 nm in width and 10 nm in length on average. A clear curved strip (red points) with a 20° bent angle ($\Theta$) can be observed, confirming our hypothesis that chiral defects (screw dislocations) may exist alone in nanocrystals. The identification of screw dislocations refers to Yurri Gunko's work on screw-dislocation-induced intrinsic chirality in CdSe/ZnS QDs[37]. The Burgers vector was measured to be 0.4 nm and was utilized for further quantum mechanics calculations. The CD strength of the IPNC sample is approximately one-tenth of the strength of traditional organic–inorganic hybrid chiral perovskites[21,25]. Considering the TEM images, only one in the 20–30 nanocrystals was found to have a dislocation, which may be attributed to the infrequent appearance of screw dislocations in nanocrystals. This gives additional evidence that the intrinsic chirality should be attributed to chiral defects. According to the phase information, a nanocrystal model with screw dislocation is shown in Fig. 3d, e. The cubic-phase CsPbBr₃ crystal is made up of PbBr₆⁻ octahedral structures (drawn in pink) and Cs⁺ ions. Stress along the shear interface resulted in screw dislocation perpendicular to the Burgers vector. The curved lattice strips in the TEM images (red points) are also highlighted in red along the (200) phase. A spiral Pb–Br line was drawn with the atoms amplified in Fig. 3e. Referring to the orbital distribution in the CsPbBr₃ nanocluster[38], the locations of the highest occupied molecular orbital (HOMO) and lowest unoccupied molecular orbital (LUMO) around Pb and Br atoms

indicate that the screw dislocation among Pb–Br crystals may result in significant optical activity. By this inference, the origins of the CD signal come from the screw orbital geometry-induced electron scattering, as discussed in many studies[39–41].

**Inversion of CD signal**. To date, MAOSIC has achieved chirality in IPNCs. However, the optical rotation direction of the synthesized samples was highly random. Precise control over the enantiomeric synthesis of chiral IPNCs is challenging due to the absence of enantiomeric chiral ligands. The adjustment of [*T*, *C*] still did not obtain the expected laevogyrate or dextral IPNCs in a few loops. To address this challenge, we referred to an organic synthesis method that inverses the molecule chirality through temperature and orientation adjustments[42]. Dislocation formation is a nonequilibrium process[43], in which light induction has a greater effect than temperature[44]. We applied a femtosecond pulse laser (800 nm, 1000 Hz, 150 fs pulse width) to introduce thermal vibration in IPNCs, and we found chirality inversion events. As shown in Fig. 4a, after 300 s of irradiation in solution, the CD signal of the local optimal sample near the first exciton absorption peak showed inversion from L to D. The same results were achieved for chirality inversion (Fig. 4b) in the global optimal sample (from the original D to L). The peak wavelength

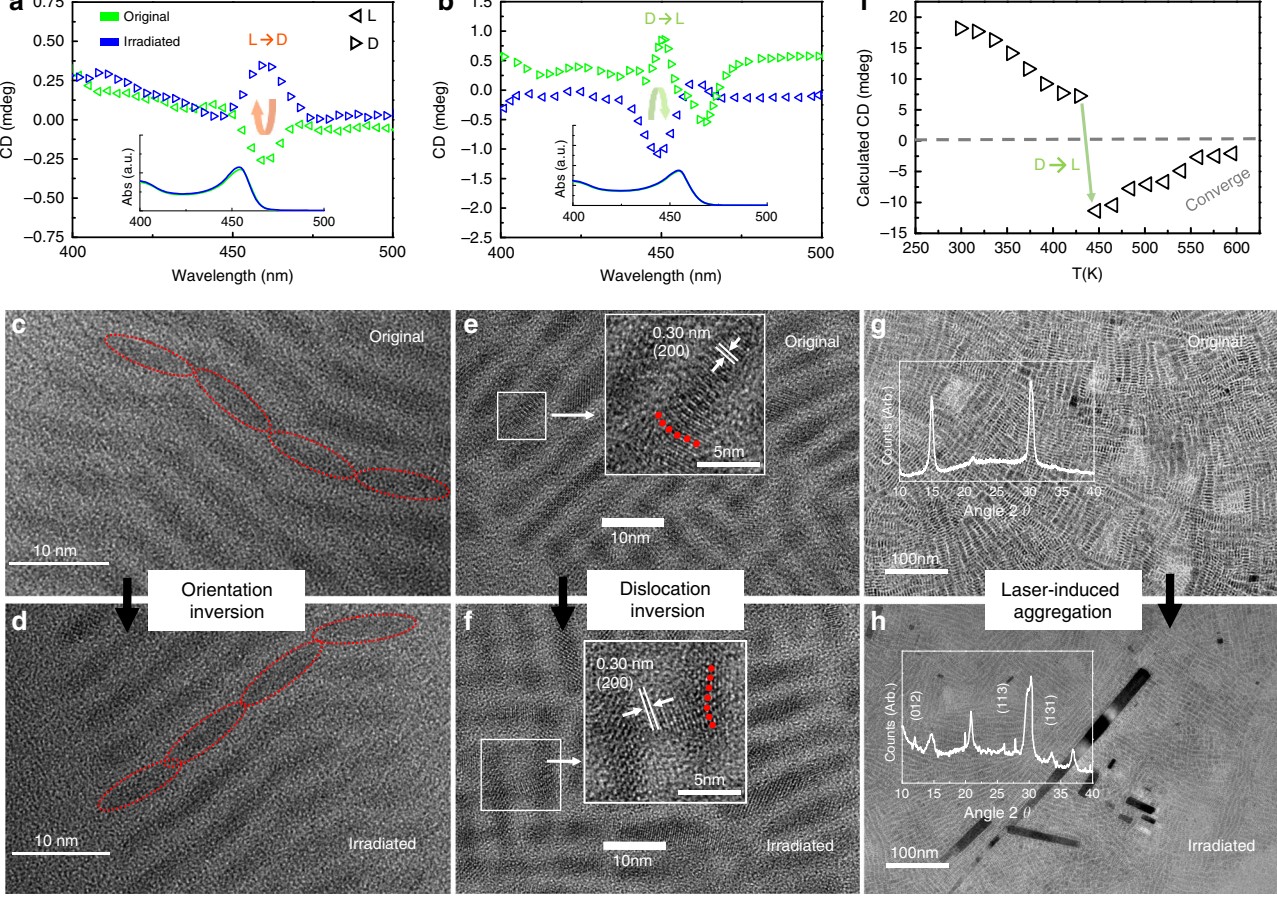

**Fig. 4 Investigation of the laser-induced chirality inversion of IPNCs.** Laser-induced chirality inversion was achieved both in **a** the local optimal sample (from L to R) and **b** the global optimal sample (from R to L). An inversion of orientation was observed by comparing the TEM image of **c** the original local optimal sample and **d** that after laser heating. Evidence of dislocation inversion is also captured in the HRTEM image of **e** the global optimal sample and **f** the sample after irradiation. **g**, **h** Aggregation occurred after laser treatment. The XRD patterns were inserted for comparison. **i** Numerical data with the parameters observed from the experiments.

and intensity of optical absorption varied by <1% (the inserts in Fig. 4a, b), and the CD intensity after irradiation was similar to the original intensity. Combined with the TEM images (Fig. 4c–f), the shape and size of the nanoplates remained unchanged during the laser treatment. In particular, the reason for the chirality inversion of these two samples might be different. According to the TEM images of the local optimal sample (Fig. 4c, d), the chirality inversion was regarded as belonging to the orientation inversion of the chiral assembly. Similar to reported works[45,46], this phenomenon verifies that light radiation of a solution of nanocrystals enables the formation of chiral crystals with optical chirality. However, for the global optimal sample (Fig. 4e, f), the inversed screw dislocation direction should be the key issue. We attribute both of these kinds of inversions to the laser-induced thermal vibration in the nanostructure. The mechanism of thermal-induced chirality inversion is discussed in detail in the next section and provides further evidence of screw dislocation in chiral nanocrystals. Similar treatment under a 300 nm wavelength femtosecond laser was also applied to both samples. For the 300 nm wavelength laser with less of a thermal effect than the 800-nm laser, no inverse CD signal was obtained. The transient absorption spectrum of the global optimal sample was collected. A 90% decrease in absorption at 5 ps and a 39% increase in the average carrier lifetime indicate the existence of defect-induced intermediate states after 800 nm laser irradiation (Supplementary Fig. 12). For both samples, slight aggregation occurred according

to the TEM images (Fig. 4g, h). The XRD patterns (inserts in Fig. 4g, h) were collected, and the additional peaks at 12°, along with the peaks near 20°, 26°, 28°, and 33° belonging to the XRD pattern of the $Cs_4PbBr_6$ crystal, were observed. A phase transition and regrowth process from $CsPbBr_3$ nanocrystals to $CsPbBr_3$/$Cs_4PbBr_6$ core/shell nanostructures occurred, which has been reported in many works when heating ion-rich $CsPbBr_3$ nanocrystal solutions[47,48]. The driving force of the phase transition and self-assembly was investigated to be the thermal-induced reaction with the excess amines in solution[49]. The interior crystalline $CsPbBr_3$ phase remained, which agrees well with the reported light-induced aggregation of $CsPbBr_3$ nanocrystals[50]. Furthermore, the NEB calculation (Supplementary Discussion 1) provides a feasible path from $CsPbBr_3$ to $Cs_4PbBr_6$ at the edge of the $CsPbBr_3$ nanocrystal.

## Solution of circular dichroism with the Schrodinger equation.
To describe the thermal fluctuation induced by temperature, a harmonic potential $cr^2$ is added to the Schrodinger equation to modify the equation obtained earlier by other researchers through pure dislocation analysis[51,52]. The harmonic potential here refers to the thermal oscillation, which can be denoted the same as in Einstein's model, $\hbar\omega$, for the phonon energy and is proportional to the temperature. This temperature, however, is not the traditional temperature of the surroundings. Since the inversion of the

chirality of the nanoplates was observed after laser excitation, the heat was absorbed only locally by the nanoplates. Thus, the energy converts only to its phonon oscillation.

$$-\frac{\hbar^2}{2m}\left[\frac{1}{r}\frac{\partial}{\partial r}\left(r\frac{\partial}{\partial r}\right)+\frac{1}{r^2}\frac{\partial^2}{\partial\varphi^2}+\frac{\partial^2}{\partial z^2}+\frac{2\beta}{r^2}\frac{\partial}{\partial\varphi}\frac{\partial}{\partial z}-\frac{2\beta^2}{a^2 r^2}+V(r,\varphi,z)\right]\psi$$
$$+cr^2\psi=E\psi \quad (1)$$

where $\beta=b/2\pi$, $b$ is the $z$ component of the Burgers vector, $a$ is the lattice constant of the material, and $V(r,\varphi,z)$ is the confinement potential that is equal to infinity outside the cylinder and zero inside the cylinder.

The energy of state $(n,l,k)$ is defined as

$$E_{nlk}=\frac{\hbar^2 k^2}{2m}+2\hbar\sqrt{\frac{2c}{m}}\left(\frac{\sqrt{l^2+2\beta kl+\frac{2\beta^2}{a^2}}}{2}+n-\frac{1}{2}\right) \quad (2)$$

The solution (Detailed information for solving the Schrodinger equation can be found in Supplementary Method 6.) to the Schrodinger equation can be expressed as[53]

$$\psi_{nlk}(r,\varphi,z)=N_{nlk}\,WhittakerM\left(-\frac{1}{4}\sqrt{\frac{\hbar^2}{2mc}}\left(k^2-\frac{2mE_{nlk}}{\hbar^2}\right),\frac{\sqrt{l^2+2\beta kl+\frac{2\beta^2}{a^2}}}{2},\sqrt{\frac{2mc}{\hbar^2}}r^2\right)$$
$$r^{-1}e^{i(l\varphi+kz)} \quad (3)$$

No boundary condition is available to this solution of Eq. (3) since the zeros of the WhittakerM function could not be analytically expressed. However, the solution could still be applied, and it describes the penetrating property of the screw dislocation. The solution can be numerically solved when given the parameters required and the quantum numbers. The shape of the wave function, in the region where $r$ is small, is similar to the one solved from the Schrodinger equation without the harmonic potential, while in the region where $r\rightarrow R$, the magnitude of the probability increases, and the wave function is no longer bounded by radius $R$. In this way, the boundary condition could actually be introduced into this wave function by making the coefficient in front of the harmonic potential a function of $R$, i.e., $c=c(R)$. The larger the radius is, the slower the harmonic potential should increase. Thus, $dc(R)/R<0$.

**Temperature dependence of the chiral intensity**. The model above also contains information about the temperature dependence of circular dichroism. The influence of the temperature can be seen from two parameters, $\beta$ and $\omega$, if the second-order harmonic potential $cr^2$ is set equivalent to $n\hbar\omega$ by applying the Einstein phonon model for the average frequency $\omega$. Here, $\beta$ is proportional to the magnitude of the Burgers vector $\boldsymbol{b}$, which measures the magnitude of the dislocation. Thus, it is reasonable to assume that the circular dichroism decreases together with the magnitude of $\beta$. This could be seen from our numerical calculation of circular dichroism by adopting the method introduced in previous research[52]. With the theory re-illustrated above, the numerical results have shown that the circular dichroism decreases together with $\beta$ monotonically and vanishes at $\beta=0$.

Another aspect worth noting is that the energy of the dislocation may not be symmetric. Note that the Burgers vector in the Schrodinger equation is positive, which is right handed with the partial derivative with respect to $\varphi$, i.e., if one winds their right hand in the same direction as $\varphi$, the thumb will point in the direction of $\beta$. Thus, if $\beta$ is negative, this becomes left handed. Thus, the signs of $\beta$ are actually asymmetric with each other, and their free energies differ. Since they are asymmetric, the system favors the lower energy state, while an energy barrier exists that prevents the transition. By reaching a certain temperature, the

nanoplates can provide enough energy to overcome the energy barrier and finish the transition. After the transition, since the high temperature always favors fine symmetry, in which case there is no dislocation energy and is, thus, at the lowest in energy, the dislocation slowly decays away, and the CD signal vanishes at the same time.

The intensity of the phonon oscillation introduced by $cr^2$ is also related to temperature; $\beta$ will decrease and even reverse with increasing temperature and $\omega$ will increase with temperature, making $\beta$ insignificant. The phonons obey the Bose-Einstein distribution[54], and from Einstein's model, the total energy is $n\hbar\omega=\left(e^{\hbar\omega/k_B T}-1\right)^{-1}\hbar\omega\rightarrow k_B T$ when $\hbar\omega\ll k_B T$, proportional to temperature. Thus, the phonon energy also plays a role in canceling the chiral property.

The CD intensity was then calculated to show its temperature dependence (detailed parameters and procedures are in the Methods section and Supplementary Method 7), results are shown in Fig. 4(i). Note that the value here is larger than the value observed in the experiments. This is due to the dilution of the CD signal by the nanoplates with no screw dislocation. The model described above is for a screw-dislocation-equipped cylinder, while in real cases, there is only one such nanoplates in every several dozens of screw-dislocation-free nanoplates. Thus, the CD signal measured is much smaller than that calculated. Further, the incident light in our model is from the $z$-direction of the cylindrical coordinate, while in real cases, the light is incident from arbitrary directions, which also increases the difference between the CD signal measured and our calculation. The calculation result is expected to be several dozen times higher than the experimental result, which agrees well with our calculation, where the ratio is ~20.

The reversal of chirality is due to the asymmetric-energy-state assumption made above. The reversal temperature is converted from the laser energy used in the experiments. Upon reaching that energy barrier, the nanoplates acquire enough energy to move to the lower energy state, whose dislocation is reversed, and the CD signal is also reversed. This coincides with the model since the CD signal reversed its sign by reversing the sign of $\beta$. Further, in each state, as the parameter for the temperature, i.e., $c$, increases, since a high temperature always favors fine symmetry, it can be seen that the circular dichroism decreases as if the parameter $\beta$ is decreasing. This is similar to the effect mentioned above; increasing $c$ in the formula causes $\beta$ to be insignificant and therefore reduces its CD. We also calculated the temperature dependence of ligand-induced chirality for the inorganic system (Supplementary Method 8). The CD signal decays with increasing temperature without reversal, which further confirms the existence of screw-dislocation or chiral assemblies in synthesized IPNCs.

**Discussion**

In summary, we successfully discovered optical chirality in CsPbBr$_3$ nanocrystals through the intelligent cloud lab. The CD signal was maximized autonomously under the guidance of optimization algorithms. Screw dislocation and chiral assemblies were captured by TEM to provide a possible explanation to the origin of chirality. Then, a quantum mechanical theory for semiconductor nanoplates with temperature-dependent chirality was presented. With the AI system in this work, we fundamentally improved the scope of these unique nanomaterials in the areas of biosensing, chiral catalysis, and chiral photonics by combining with a cloud technique. With these advances, we believe that this strategy provides not only insights into an understanding of the chiral origin of nanocrystals but also interdisciplinary links between chemistry and IT technologies. Moreover, this self-driving cloud lab frees scientists from

dangerous and tedious experimental tasks, significantly reduces the setup costs of traditional in-house labs, enhances the collaboration efficiency, and reliability among researchers, and most importantly, accelerates the discovery of new materials.

## Methods

**Software**. Materials acceleration operation system in cloud (MAOSIC) is a software package to manage the automated platform. MAOSIC cores are implemented in Python and can run on laptop CPUs with multiple operation system including Windows 10, most Linux based distributions and MacOS.

**Instruction compile**. All robotic automation and characterization modules are controlled by a hardware interface in MAOSIC, which includes both high-level and low-level instructions based on JSON-RPC2.0 protocol. In MAOSIC, the compile process indicates the language translation from high-level instructions to commands for chemical machines (Supplementary Information Section S4). An example is shown in Supplementary Fig. 9. The high-level instruction is a formula made up of instruction symbols and operators. Each symbol indicates the corresponding hardware module, and each module has its specific parameter vector, which represents all adjustable parameters this module can provide. The symbol array collected by operators can describe the arrangement of modules and reagent in both time and space domain. Five kinds of operators are utilized: "+", "×", "." and "|". By combining different module symbols and the operators, complex experiments can be demonstrated. Details of all instruction symbols, parameter vector and operators are provided in Supplementary Tables 1 and 2.

**Optimization algorithms**. The parameters for the SNOBFIT based reinforcement learning (RF) in this work is of the form like $(\mathcal{S}, \mathcal{A}, [\mathcal{P}_{as}], \mathcal{R})$. Here $\mathcal{S}$ denotes the set of state $s$. State at timestamp $i$ is denoted by $s_i = [a_0, r_0, \ldots, a_i, r_i]$; $\mathcal{A}$ denotes the set of action $a_i = \left[ a_i^1, \ldots, a_i^j, \ldots, a_i^n \right]$; $[\mathcal{P}_{as}]$ denotes the state transition probability, here $p_{as}$ decides what action $a$ to make under the experimental conditions. $\mathcal{R}$ denotes the reward function under state $\mathcal{S}$ and action $\mathcal{A}$. $r_i = \left[ r_i^1, \ldots, r_i^j, \ldots, r_i^n \right]$, the reward function denotes the mapping $r_i = R(a_i)$. At a specific experimental condition $a_i$ is mapped to get the difference between the experimental and the target then get the reward $r_i$. For a timestamp $i$ with the previous state $s_{i-1}$, the policy function decides the action to make by $a_i = \pi(s_{i-1})$. Then, continue the experiment with $a_i$, and get the reward $r_i = R(a_i)$. For the MAOSIC, architecture, the state could be updated by $s_i = s_{i-1} \cup [a_i, r_i]$. Policy function $\pi$ performs local optimizations around the best points while searching for unexplored regions to ensure global optimality. For local optimization, the action is decided by searching from the best condition with a full quadratic model. For example, $q(a) = r_{best} + g^T(a - a_{best}) + \frac{1}{2}(a - a_{best})^T G(a - a_{best})$, where $g$ and $G$ are the gradients and the symmetric matrix estimated, respectively. For the global optimization, the action is decided by searching simultaneously in several promising sub-regions. For example, the parameter space is partitioned into sub-boxes each contains a history state. The actions are chosen by

$$a = \begin{cases} \frac{1}{2}\left( \underline{a_i} + a_i \right), & \text{if } a_i - \underline{a_i} > \overline{a_i} - a_i \\ \frac{1}{2}(\overline{a_i} + a_i), & \text{otherwise} \end{cases}, \text{ where } [\underline{a_i}, \overline{a_i}] \text{ is one of the largest sub-boxes}$$

of the current partition with history action $a_i$ inside. The actions selected will finally be rounded into the nearest integral multiple of the resolution vector $\Delta a$, which is decided by the physics condition of reaction parameters (here $\Delta a = (\Delta T_i, \Delta c) = (0.5\,°C, 0.01\,mmol/mL)$).

**Robotic and microfluidic platform**. 7 DoF (degree of freedom) Robot arm (Gen 3, Kinova Inc, Canada) and an integrated gripper (2F-85, ROBOTIQ, Canada) is utilized for automated transfer. The reaction temperature adjustment was realized through a copper pipe (OD = 20 mm, $L$ = 150 mm), an electric heating cartridge, a temperature sensor and a PID controller. PTFE tube was coiled and sat within a groove with a 1-mm radius engraved on the surface of the copper pipe. The heating cartridge ($D$ = 14 mm, $L$ = 150; mm, Max power = 100 W) was embedded in copper pipe. The real temperature was monitored by a patch K-type thermocouple ($-100\,°C \sim 350\,°C$) with 0.1 °C accuracy, embedded between PTFE tube and cooper. The temperature was controlled by a PID controller, with a temperature variation below ±1 °C after 5-min warm-up. The PID device was connected through serial communication by STM32 microchip. OEM (Original Equipment Manufacturer) syringe pumps (MSP1-C2, Longer, China) were used to inject the dispersed phase (Cs precursors, Pb precursor, and ODE) toward a polyether ether ketone (PEEK) cross-junction to mix and into the heating region. 0.5-ml glass syringes were used. The injection cross-junction and the syringes carrying the precursor solutions were connected through PTFE tubing (ID 1/32″, OD 1/16″) using PEEK finger-tight fittings. The mixed precursor solution flows through PTFE tubing (ID 1/32″, OD 1/16″) coiled around heating cartridge. The reaction time was controlled by varying the total flow rates of two kinds of precursors.

**Preparation of Cesium oleate**. Cesium oleate precursor solution. $Cs_2CO_3$ (0.815 g), ODE (40 mL) and OA (2 mL) were loaded in a 100 ml three-neck flask, dried at 120 °C under vacuum for 2 h to dissolve the cesium salt and to dry the solution. The solution was then cooled, stored in bottle and collected with glass syringe P1 through PTFE tube.

**Preparation of lead precursors**. $PbBr_2$ (0.089 g) powder were added in a 20-mL flask together with dried ODE (5 mL). The mixture was then dried under vacuum for 2 h at 120 °C. After 2 h, 1 mL of dried OA and dried OLA were added under argon until all $PbBr_2$ dissolved completely. The solution was then cooled, stored in bottle and collected with glass syringe P2 through PTFE tube.

**Synthesis of $CsPbBr_3$ IPNCs**. The injection rate of syringe pumps: P1 (Cs precursors), P2 (Pb precursors), and P3 (ODE) were controlled to adjust the parameter $C$ (mmol ml$^{-1}$), which represents the molar concentration of $Cs^+$ ion. In all experiments, P1/P2 was set as 0.5 for a lead-rich synthesis. On the other hand, the sum of P1, P2, and P3 was kept unchanged to ensure the same reaction (heating) time in all loops of experiments. The variation of reaction temperature was set within [90 °C, 180 °C].

**Optical characterization**. Deuterium-Tungsten Halogen light source (200–2500-nm output, DH-mini, Ocean Optics, UK) was used for inline absorption measurement. The illumination light was collimated through an optical collimator (F230SMA-C, Thorlabs, Germany) and a long working distance object (×10, Mitutoyo). A plano-convex cylindrical lens (GCL-110115, Daheng Optics, China) was used to shape the light into a line (1-cm length) along the direction of tube. The transmission light was collected by spectrometer (USB2000 +, Ocean Optics) through a multimode optical fiber. CD measurements were conducted on a JASCO J-1500 CD spectrometer. The scan rate was 20 nm/min. All CD experiments were carried out in Milli-Q water with a quartz cuvette (0.1-cm path length, Hellma). fs-TA spectroscopy is based on a Ti:sapphire regenerative amplifier system (Spectra-Physics, Inc.) that produces 800 nm pulses with a 100 fs pulse width and 1000 Hz repetition rate. The 800 nm pulses were irradiated onto a $CaF_2$ crystal to generate white probe light (360–750 nm). In order to prevent $CaF_2$ crystal from being damaged by the laser, it was moving horizontally at low speed. The pump-induced changes in the transmission ($\Delta T/T$) of the probe beam were monitored using a monochromator/photomultiplier configuration with a lock-in detection.

**Crystalline characterization**. Bruker Advance D8 Ew (Germany) X-ray diffraction (XRD) equipment is utilized for identify the phase of synthesized IPNCs. The TEM and HRTEM images of IPNCs is characterized by FEI Tecnai G2 F30 under 300 kV, 120 μA. The purification and TEM sample: Mix with ethyl acetate with 1:1 ratio and centrifuge at 8000 rmp for 4 min. Take the deposit sample and dissolved in 5-ml n-hexane. Centrifuge for 2 min at 4000 rpm. Deposit the solution on ultrathin carbon grid for natural volatilization.

**DFT simulation**. We use the Perdew−Burke−Ernzerhof (PBE)[55] exchange correlation functional in Quantum Espresso code[56]. A 60 Ry kinetic energy cut-off and gamma K-point are applied for the ground state calculation.

**Calculation of circular dichroism**. The calculation follows the formulae in Supplementary Information S9, which calculates the overlapping of different chiral states solved from the Schrödinger equation (Supplementary Information Section S8). It was carried out with experimentally observed values for the parameters: $|\boldsymbol{b}| = 0.4$ nm, $R = 2$ nm, $L = 5$ nm. Since the penetration depth of the circularly polarized light was measured to be ∼ 1 nm, and the absorption peak is approximately $\omega \sim 10^{15}$ Hz, the imaginary part of the refractive index will reach ∼ 100. This follows from the attenuation constant from the Beer-Lambert Law[57,58]. $I = I_0 e^{-\alpha x}$. The penetration depth is defined as $\delta_p = 1/\alpha$, and $\alpha = \frac{\Im(n_{complex})\omega}{c_{light}}$, where $c_{light}$ is the speed of light. Adapting these parameters into the calculation of $\Gamma(\omega)$ in Supplementary Information Section S9, the temperature dependence of the CD intensity can be calculated. Assumptions made for the calculations are that the excitation takes place only from the ground state to the first excited state, i.e., from $(n, l, k) = (1, 0, 0)$ to $(n, l, k) = (1, \pm 1, k)$, and the absorption peak $\omega$ is the same as that at 273 K over the entire temperature range (assuming barely no change to the absorption peak during this process, which agrees with the experiments). The resulting plot is shown in Fig. 4(i), where the horizontal axis is the equivalent temperature adopted from Einstein's model by equating $k_B T$ to the energy of the harmonic oscillators, which is proportional to the square of the displacement of the lattice, which is ~1 Å, and the vertical axis is the exact CD(mdeg).

## Data availability

The data that support the findings of this study are available from the corresponding author upon reasonable request.

## Code availability

The code of MAOSIC is now continually updated to GitHub. Custom code to run experiments was closely integrated with the experimental setup and hard be executed without associated hardware. Interested readers can contact corresponding author [zhuxi@cuhk.edu.cn] for website address and extract the code.

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

## Acknowledgements
This work is supported by the Shenzhen Fundamental Research Foundation (JCYJ20170818103918295, JCYJ20180508162801893, JCYJ20190808121211510), and the National Natural Science Foundation of China (grant no. 21805234). Also supported by funding (2019-INT018, 2020-IND002) from Shenzhen Institute of Artificial Intelligence and Robotics for Societ (AIRS).

## Author contributions
T.H., J.C., and X.Z. conceived idea and concepts, and initiated the research. J.-G.L. and Y.T. constructed the cloud lab platform and MAOSIC. J.-Z.L. and Y.L. remotely tested the lab for IPNCs synthesis. J-Z.L. and T.H. conducted TEM measurements and optical characterization data analysis. R.L. and X.Z. conceived the theoretical concepts and conducted the quantum mechanics calculation and DFT simulation. J.-G.L. and X.Z. prepared the Figures. J.C., J.-G.L., T.H., and X.Z. wrote the paper with input from all authors.

## Competing interests
The authors declare no competing interests.
