## [Peer Review File · Nature Communications]

Reviewers' comments:

Reviewer #1 (Remarks to the Author):

Authors report a 'cloud based' platform for the optimisation of synthesis conditions for chiral perovskites. They provide a comprehensive theoretical analysis to explain the mechanism behind the origin of chirality by modelling screw dislocation. 3

Overall, it is not convincing that the robotic platform accelerated the material's discovery. Furthermore, the 'cloud' aspect of the work seems tacked on, and it is not shown how the cloud aspect of the work contributes to accelerated discovery.

Generally, while the cloud aspect of the work is of broad appeal, it is not convincing that it really contributed anything to the discovery. Conversely, while the materials aspect of the work may be more convincing, it is not applicable to the broader scientific community.

The authors explore a 2D search space (concentration vs temperature) using the SNOBFIT algorithm and 250 experiment loops. Given that this undoubtedly translates to a great number of experiments overall, and considering the rather simple search space, I wonder if the authors' algorithm is any better than a random search or even a human-driven (manual) search. These experiments should be performed for comparison.

DFT calculations are performed but no detail is given about how these were done. From the figure it seems that the authors used a cluster model to define the perovskite. If so, this requires justification.

Overall, I cannot recommend publication.

Reviewer #2 (Remarks to the Author):

The authors have written an article describing the development of a fully automated robotic setup using algorithms to produce high throughput research, testing a wide range of preparation conditions for CsPbBr₃ nanoparticles.

They have built a setup from scratch, incorporating a range of preparation and characterisation techniques to form a fully automated preparation and characterisation workflow which can be controlled remotely. The setup is able to control the reaction temperature and precursor concentration with very high accuracy, and a feedback algorithm allows the system to iteratively test out different parameter combinations in order to study a range of properties.

In this case, the authors record the absorption spectrum and the circular dichroism (CD) and find a preparation condition with a high degree of circular dichroism at around 130 °C and 0.8 mmol/mL precursor solution. They then characterise the nanoparticle produced at this condition further using high resolution TEM (HR TEM) and claim to observe the presence of screw dislocations in the nanoparticles, to which they attribute the circular dichroism observed. Finally, they induce changes in the polarisation direction of the nanoparticles by laser pulsing, and set up a quantum mechanical model describing the change in dichroism as a function of temperature.

The combination of robotics and iterative AI use for material synthesis and characterisation is very impressive, and the setup is an engineering feat and could form the basis of much future material development and characterisation. The ability to test hundreds of parameter combinations in a short period of time is very promising and has the potential to fundamentally change how these materials are studied.

Similarly, the ability to measure absorption and circular dichroism on the fly, and using the information to prepare the next solution, effectively honing in on an optimal recipe is impressive, and has allowed to authors to synthesise a compound with a high degree of circular dichroism, a

phenomenon which is not widely seen in the perovskites usually studied today.

The inorganic perovskite nanocrystals typically showing CD usually have chiral ligands or organic molecules associated with them, but the authors claim to synthesise fully inorganic nanoparticles with no ligands, which still show CD. They attribute this to the presence of screw dislocations, as theorised would be possible in inorganic materials, and shown in highly chiral nanowires in refs. 27 and 28, respectively. The authors claim to observe a same kind of screw dislocation-induced chirality.

The paper is overall quite well written, and only needs minor editing for typos and clarifications. The sections about the automated setup are in my view exceptional, although I do admit not to have a strong background in robotics and automation. I also do not have a strong background in quantum mechanics, and I hope another reviewer will be qualified to assess this work.

The sections related to the HR TEM results do, however, require some further analysis and/or explanations. The presence of CD is an interesting property, but there are a range of questions that must be addressed regarding these results and their interpretation before the paper can be considered for publication in Nature Communications.

Below are some general points, followed by text-specific points with references to the line number in the pdf version.

- The evidence for screw dislocations in the nanowires is not clear at best, and if the dislocations were the origin of the chirality, their random distribution would intuitively cancel out any local chirality. The authors have drawn red dotted lines in Figure 3, which are supposed to represent screw dislocations, but the lines seem to follow the surface of the nanoparticle more than anything, as seen by the amorphous region (probably the carbon grid) seen next to the supposed dislocations. If the nanoparticles contained screw dislocations, they would be expected to be found within the crystal structure itself, and not at the edges of it.
- The authors apply pulsed laser on the nanoparticles to induce thermal vibrations in the nanoparticles, and observe a change in chirality as a result. It is not clear what the exact purpose of the pulsed laser is, other than thermal activation, in which case it should be stated which temperature the particles ended up at. The laser treated particles experience a change in chirality, which the authors explain with the quantum mechanical model, showing that chirality can change as a function of temperature, with certain temperatures corresponding to certain inversions. If the films were all treated at the same temperature, one would expect that only one chirality change would be possible (for example, at 75 °C D->L would be possible, but not L->D, according to Figure 4 (d)). If this is the case, only one type of inversion should be observed, but the authors observe inversions in both directions. This should be better explained in particular, and the chirality change could be better explained in general.
- A significant change in morphology is seen after the laser treatment, seen in Figure 4 (c). The authors state that the overall phase is the same after the treatment, namely the cubic CsPbBr₃ phase, with small amounts of Cs₄PbBr₆ present on the surface of the newly formed particles, as evidenced by XRD. The new morphology seems so fundamentally different to the original one that this claim is hard to understand. There are quite clearly at least two different types of nanoparticles present, but it is difficult to impossible to examine them in detail in the current micrograph. The smaller particles do not show the same fringes as they did in Figure 2 (a) before treatment, so it is possible that they have amorphised as a result of the laser treatment. The new phases after laser treatment must be properly accounted for and described. If, for instance, it is the Cs₄PbBr₆ compound that displays chirality, the findings do not pertain to perovskite CsPbBr₃ nanoparticles.
- There is some important experimental information missing, such as the microscope conditions used (including microscope type, voltage, dose rate, etc.) and the TEM sample preparation methods. How were the nanocrystals deposited on the TEM grids?

Following are some text-specific comments:

64: It would be useful if MAOSIC is defined in the main text as well as/instead of in the abstract.

83: 'sent' should probably be 'send'.

Figure 1: What happened to the spent materials?

Figure 2: Are (b,c) measured at room temperature? Which synthesis temperature(s) was/were

used? Units on the concentration legends would be useful.

130-132: Which is the first excitonic peak? Why only the first one? 0.8 mmol/mL has both a positive and a negative peak, which one was used? If the positive peak, how can it be compared to the other, negative ones. If the negative one, what does the positive mean in this case?

142: 'crystallinities' should probably be 'crystal quality'.

148: What is a 'cotton effect'? How does it show?

148-150: This sentence is not clear and could use some rewording and a reference.

156: 'HUMO' should probably be 'HOMO'.

Figure 3: There seems to be a shoulder to the left of the (100) peak, what does that correspond to?

(c) The simulation seems to be done on the orthorhombic phase since the Pb-Br bonds are bending? Why is this? Does this affect the simulation results?

Is the simulation done on four unit cells only? It seems odd that the LUMO/HOMO would be localized in each end of the box, and not distributed symmetrically. Why is this?

(d) What are the experimental conditions these images were taken under? What dose rate and total dose was used? How were the crystals transferred to the TEM grids? The red dots drawn on the figure seem to follow the surface/edge of individual nanorods and not screw dislocations within the crystals. There seem to be some kind of defects/dislocations just above the scale bar on the left and between the two red dotted lines on the right, but it is difficult to verify if these are screw dislocations from the given image. Is there other evidence for the presence of screw dislocations? If screw dislocations are present in both directions, intuitively no overall chirality expected.

166-169: There is no justification in the text or reference explaining why the supposed locations of the HOMO and LUMO would somehow connect screw dislocations with optical activity in the nanocrystal.

170-171: Where are the vacancies and self-assembled screw dislocations shown in the document?

171-172: Is there any evidence or reference for this claim?

174: Where is this claim illustrated or supported?

174-175: Does this mean that it was either left or right polarised within an entire sample, but it was difficult to control exactly which one of the polarisations came out, or that the polarisation overall was random within a sample?

178-180: The previous sentence stated that no overall chirality was possible to observe. How can the pulse inverse it if it cannot be distinguished initially?

Why was laser used and not just thermal heating?

181: It would probably be good to define what 'L' and 'D' refer to. It is implied due to the chirality discussion, but not explicit.

181: Which figure does 'The insert picture' refer to? Figure 4 (a) and (b)? It should be stated explicitly.

Figure 4: (a-b) The inversions happen at different wavelengths, why would this be? Are these different concentrations? If so, which?

Why do both the blue and green data cross the 0.0 mdeg in (b)?

What temperature was achieved during the laser treatment? If the same temperature was achieved, there should only be one chirality inversion observed as per (d).

(c) The nanoparticles here do not look anything like the ones seen in Figure 3 (a,d). Where is the evidence that the edges are Cs₄PbBr₆ and everything else is CsPbBr₃? The XRD pattern changes more than just the appearance of a few small peaks. The big peak around 30 degrees seems to be two peaks on top of each other and a very sharp, small peak around 20 degrees is present. Are these consistent with Cs₄PbBr₆? The ordering in the larger rods seems visible at the lower magnification, but not in the smaller rods. Are these still crystalline, or have they amorphised? The y-axes of (a,b) are 'CD' but in (d) it is 'CD strength'. Is this on purpose? If so, what is the difference?

277: The units given in the figure are mdeg. If they are arbitrary, they must be normalised relative to some reference value.

278-279: Is this preparation temperature or experimental temperature?

Reviewer #3 (Remarks to the Author):

In general the grammar needs improvement, in some cases the grammar impedes understanding of the paper. The authors should also be careful to define all the acronyms used in the manuscript.

There are a few claims about the material produced that are not well-supported. First, based off the TEM images in Figure 3, the authors claim platelets are produced, but give no evidence to differentiate the material from nanorods. Also, in their mathematical model a nanorod is assumed. The presence of platelets is seen in Figure 4, but here the image is of an irradiated sample, showing signs of degradation. So my overall question after reading the first section is whether the synthesis has produced platelets or rods?

After irradiation, there are platelets in the TEM image and peaks from Cs₄PbBr₆ (lines 186-188) in the XRD pattern. It seems likely that the irradiation is doing a lot more than switching chirality. How does the degradation of the nanocrystal material (whether it be platelets or rods) impact your measurement of chirality or the analysis of the model?

The claim that the Cs₄PbBr₆ is only on the "edge of aggregated plates" (line 187) is completely unfounded. Is there any evidence that just the edge of platelets would be phase changing, as opposed to whole nanocrystals under irradiation?

In Figure 4b, it looks like the "original" sample has a mixture of both D and L. Is there any control over which chirality is produced by the synthesis? Or do you produce a mixture in each batch? Can you separate them post-synthesis if you produce both? Please comment on the composition of the as-synthesized sample.

In the introduction (line 54) a phase transitions are listed. These are material-dependent and can't be quoted for ABX₃ perovskites in general. The authors should specify which material they looked up for the stated phase transition.

On line 172, Figure 4e and 4f are cited, but do not exist.

Overall, the idea of using AI to synthesize new materials is interesting, but this work needs major revisions before being considered for publication.

Reviewer #1 (Remarks to the Author):

Authors report a ‘cloud based’ platform for the optimization of synthesis conditions for chiral perovskites. They provide a comprehensive theoretical analysis to explain the mechanism behind the origin of chirality by modelling screw dislocation.

Comment 1-1: Overall, it is not convincing that the robotic platform accelerated the material’s discovery. Furthermore, the ‘cloud’ aspect of the work seems tacked on, and it is not shown how the cloud aspect of the work contributes to accelerated discovery. Generally, while the cloud aspect of the work is of broad appeal, it is not convincing that it really contributed anything to the discovery. Conversely, while the materials aspect of the work may be more convincing, it is not applicable to the broader scientific community.

Reply 1-1: Thanks for the comments from Reviewer 1. We find it is true that the description of the robotic platform and cloud-based operation system in this manuscript was not detailed enough, which may cause misunderstanding of this ‘cloud synthesis system’, especially for readers not in the field of automatic chemistry. Generally, the advantages of robotic automation in lab reflect the following aspects including high-throughput synthesis and screening, working in extreme environment, high reproducibility and real-time sensing. The discussion on how these chemical machines benefit the materials science research and accelerate the discovery of new materials were well summarized in recent review articles^{1, 2, 3, 4}. Detailed in this work, the hardware modules of the robotic platform were constructed by programmable syringe pumps, flow reactors, collaborative robot, and automatic characterization equipment, and all controlled by the designed cloud operation system MAOSIC. MAOSIC was developed based on a Platform as a Service (PaaS) cloud, referring to the ChemOS⁵ (Guzik et al) and the cloud synthesis system developed by Steven Ley’s group⁶. The contribution of the cloud automation platform for optically active perovskite discovery can be summarized as follow:

- (1) Scientists no longer need purchase of expensive equipment, been trapped in typical confined environments, which costs a lot on both research funding and time. The total cost of all facilities in this work is close to 2 million dollars, however, the remote researchers need only several thousand for most experiments. Furthermore, the open collaboration will be promoted and research resources are maximized to accelerate the discovery efficiency.

- (2) With the 5-9 orders of magnitude smaller than traditional bench-top chemistry, the experimental cost and space occupation based on microfluidic reactors are significantly reduced. Especially for informatics chemical synthesis, it becomes a much better choice. In such a small volume, the homogeneity of reactions, response rate to the condition changes, and energy saving can all be well improved.⁷ In this work, the high heating efficiency of reagents (heating from 25 °C to 120 °C within less than 1s, Figure S7) saves time during temperature adjustment. Additionally, the in-situ absorption test saved 90% time cost on sample preparation, data processing, and analysis. Furthermore, in this integrated platform, serials experiment tasks were divided and assigned to individual machines for parallel running, for example, the second loop of synthesis can be started while the sampling and characterization of the first loop are still running. This is far beyond the working efficiency of single labor. We have compared the reagent and time cost with traditional bench-top synthesis handled by human labor, 75% reagent cost and 40% time cost were reduced.

- (3) The robotic platform can be easily coupled with AI algorithms and achieve an autonomous exploration. In this work, the SNOBFIT-based reinforcement learning scheme was utilized to search and optimize the optical activity of IPNCs. The detailed statement of this algorithm will be discussed in **Reply 1-2**.

In fact, the field of chiral inorganic nanostructures, as an emerging area, is rapidly

expanding during the last decade. Other than well-established chirality transfers from bioorganic molecules, achiral NPs with chiral assemblies together with intrinsically chiral NPs, like the chiral perovskite discussed in this article, have revolutionized the motif of imparting handedness of nanoscale matter by means of traditional wet-chemistry, three-dimensional lithography, CVD and so forth. Moreover, with the AI system mentioned in work, we fundamentally improved the scope of such unique nanomaterials in the area of bio-sensing, chiral catalysis, and chiral photonics with a combination with mordent cloud technique. With these advances, we do believe such strategy would provide not only incisive insights for understanding the chiral origin of the NPs, but also interdisciplinary links for mordent chemistry to IT technology.

We revised Figure 1a and added a detailed description in supporting information for readers to easily understand the cloud-based system. We set up a Software as a Service (SaaS) to allow remote users designing experiments and analysis. Users no longer need to consider how to control remote equipment and worried about how to collect and analysis experiment data. With the help of the cloud system, experiments will be handled automatically, and the user can see the stored experiment data at any time.

Revised Figure 1a. Workflow and communication scheme of the cloud lab for discovering new optically active IPNCs.

For cloud software, security is an unavoidable issue. A cloud system may receive the attack from around the world. We have taken a series of measures to protect the system and user's data from public attacks, including customization of specific firewall rules that only allow data transmission with our cloud server, encryption of data transmission between local equipment, users, and cloud server, and encryption of all experiment logs stored in the cloud server and domestic machines. We adopted multiple communication encryption and firewalls to ensure data transmission security. The university firewall protects the local and equipment, and the university security policy will block all unauthorized external access. An encrypted tunnel is built between local equipment and the cloud server, which adopt transport layer security (TLS) to encrypt all data transmission. The cloud server we used provides distributed denial of service (DDoS) prevention, which ensures the availability of our system. All the users need to use the key build Socket Shell (SSH) tunnel to connect with the cloud server, so any unauthorized user has no chance to access the cloud server. For the data storage security, in both cloud server and local equipment, we adopt Linux

unified key setup to encrypt all files. Even if our data breach, without the key, the hackers are almost impossible to obtain experiment details.

Corresponding revision in supporting information are listed below:

Revised S1. MAOSIC architecture

Figure S1. Scheme of MAOSIC. Five modules were designed and intercalated with each other for autonomous materials discovery.

Figure S2. The workflow of MAOSIC.

MAOSIC runs as a web server that receives synthesis request and optimizes the corresponding reaction parameters on the first-in-first-out (FIFO) principle. Figure S1 shows the Workflow of how MAOSIC handling the request.

Environment Setup

1. Prerequisites

Hardware Requirements

- A powerful CPU
 - Laptop CPUs can often run the software, and the performance will influence software speeds
- **Our material synthesis hardware**
 - MAOSIC has a hardware control interface, which controls all hardware and translates hardware-independent instructions into elementary basic operations.

- Without our hardware, MAOSIC cannot communicate with hardware and raise exceptions.

Supported operating systems

- MAOSIC could run on multiple Operating Systems including Windows 10, most Linux based distributions, and MacOS
- MAOSIC cores are implemented in Python. So, MAOSIC could run on the most mainstream operating system

2. Install Guide

Installing dependencies

- Python ≥ 3.5 64-bit
 - apt/yum install python3 (Linux)
 - Installer (Windows)
 - brew install python3 (MacOS)
 - Python 3 Anaconda environment is recommended, which could be download from <https://www.anaconda.com/download/>
- MySQL (optional)
 - MAOSIC store long-term information through SQLAlchemy, which support multiple mainstream SQL database. MySQL, Postgres, Oracle, and SQLite are supported in MAOSIC and needs to be configured.
 - One of the databases must be configured, and MySQL is recommended.
 - MySQL could be downloaded from <https://www.mysql.com/downloads/>

Getting the MAOSIC code

MAOSIC can be directly downloaded from GitHub. For now, extract the code to a directory where you are comfortable working with it. Navigate to it with the command line.

Setup

Enter the folder that MAOSIC has been downloaded to and run the command to install Python dependent packages.

```
pip install -r requirements.txt
```

Notes

For now, this guide is far from complete. The new functionality may be updated over time, and new dependencies will be added and removed accordingly in the future.

3. MAOSIC Illustration

Overview

MAOSIC has multiple modules. These modules could run on a computer or multiple computers with internet access.

- **Hardware interface:** Control all robotics hardware, handle all hardware dependent information and provide hardware control interface
- **Cloud:** Including core of MAOSIC (handle the request and optimize the reaction parameters to meet the demand), a web server for the user to submit demand and SQL database (optional).
- **User control:** Provide multiple ways for users to interact with MAOSIC

Hardware interface

- **Arduino:** An Arduino Uno board is used to control a stepper motor and two gas valves. MsTimer2 is used to control the time accurately.
- **STM32:** A STM32 F103 development board is used to control the peristaltic pumps and a stepper motor.

- **PC:** Hardware interface communicates with microcontrollers through serial and handles all hardware-dependent information. With our robotics hardware, the hardware interface could work normally. Run `cd hardware_interface/PC & python hardware_control_interface.py`, input the port of each serial and a JSON-RPC based control interface would start on port 8888.

Cloud

Three modules (MAOSIC core, a web server and MySQL) could be distributed run on the cloud server.

MAOSIC core is fully implemented in Python, which could handle the synthesis request, control hardware to do experiments through a hardware control interface, analyze the experiment results, and learn from the results. This module automates the synthesis reaction parameter optimization after a request. The request is implemented base on JSON-RPC protocol.

MAOSIC core includes optimizer and analyzer. For the learning policy of optimizer, an algorithm based on SnobFit is used currently. For analyzer, multi-peak Gaussian wave fitting is implemented to analyze the spectrum data.

MAOSIC core needs to fetch and store data from the database management system (DBMS) so that the core could work normally. Run `python run.py` to start MAOSIC core. The request handler would start on port 9999.

A basic version website where the user could submit synthesis demand is implemented in Flask. The web server would start on port 80 after run `cd cloud\website-basic & python run.py`. This website needs to connect the database to store long-term storage.

MAOSIC core should be working when submitting the request.

User control

- **UI:** A simple UI implemented with PyQt is provided in `\user_control\UI`, which could support low-level control of the hardware. Currently, syringe pump control is supported. Open the UI by running `cd user_control\UI & python ui.py`

- **Browser:**

Users could connect the website through the browser to submit their demands.

Database Config

MAOSIC fetches and store important information from the database through SQLAlchemy. So, MAOSIC supports multiple mainstream SQL databases such as MySQL, Postgres, Oracle, and SQLite. `maosic__init__.py` records the database config URI. This database connection URI should be modified (<https://flask-sqlalchemy.palletsprojects.com/en/2.x/config/#connection-uri-format>) to configure the database.

After configuration, run `cd scripts & python create_db.py` to initialize the database table and insert little test data. Currently, we do not upload all the data in our database.

The database mainly stores the data during experiments. MAOSIC could learn from those data.

Notes

It is the first version of MAOSIC; more functionalities are under construction.

Specification of instruction symbols in MAOSIC

All robotic automation and characterization modules are controlled by a hardware interface in MAOSIC, which includes both high-level and low-level instructions based on JSON-RPC2.0 protocol. The high-level instruction is a formula made up of instruction symbols and operators. Each symbol indicates the corresponding hardware module, and each module has its specific

parameter vector, which represents all adjustable parameters this module can provide. Taking an example for the *Heating module*, the instruction symbol is H, and the parameter vector is $[T, \dot{T}, t^H]$. Here, T is the stable temperature controlled by H, \dot{T} is the temperature gradient (heating rate), t^H is the working time of H. The symbol array collected by operators can describe the arrangement of modules and reagent in both the time and space domain. Five kinds of operators are utilized: "+", "×", "·" and "|". By combining different module symbols and the operators, a language is developed for the communication between MAOSIC and the real experiments. Details of all instruction symbols, parameter vector (Table S1) and operators (Table S2) can be found in the Methods and Materials section. With the help of the language, all experimental operations can be interpreted by a formula and then convert into the machine code, which is defined as the process of “compile” (details are shown in supporting information S4, Figure S4). All parameters in an experiment will be stored in vectors to construct the parameter space, which is called “parameterization”. These vectors are input data for both hardware control interface and AI optimizer.

Table S1. Specification of instruction symbols and parameter vectors in MAOSIC

Symbol	Module	Parameters	Parameter vector
H	Heating module	temperature, temperature gradient, time	$[T, \dot{T}, t^H]$
ABS-CD	Abs-CD Measurement	integration time	$[Ip]$
CR	Collaborative robot	initial point, final point, time	$[P_i, P_f, tr]$
SP	Syringe pump	injection volume, time, injection rate, times of injection	$[vi, ti, \eta, \tau]$

Table S1 symbolizes and parametrizes all the modules in MAOSIC, with time dependence.

We define θ as the vector that concludes all adjustable parameters as:

$$\theta = [T, \dot{T}, t^H, Ip, vi, ti, \eta, \tau, P_i, P_f, tr]$$

The θ vector delivers orders to all the modules in MAOSIC for the given project, and it also guarantees the repeatability for the typical task. The materials, like reagents, are also parametrized to make the whole system digitalized. We define R_i the liquid reagent, as $R_i = [i, v_i]$. R_i^s the solid reagent, as $R_i^s = [i, w_i]$. Here v is the volume of liquid reagent, w is the weight of solid reagent, i is the serial number of the reagent in database. MAOSIC conduct the experimental operation by combining the parameter spaces of modules and the chemical reagent, and an operator is essential to connect the two parameters space. As shown in Table S2, there are four well-defined operator symbols with the representation and examples, and each operator correlates the symbolic calculations within the module and chemical parameters. For example, $R_1 \cdot SP_1$ can be interpreted as MAOSIC transfer the liquid chemical R_1 by the first Syringe pump, as indexed SP_1 from Table S1.

Table S2. Meaning and usage of operators.

Operator symbol	Representation	Usage example
\cdot	Reagent transfer	$R_1 \cdot SP_1$ (transfer R_1 with SP_1)
$+$	Fusion of reagent	$R_1 + R_2$ (Adding R_2 into R_1)
\times	React under environment	$R_1 \times H$ (R_1 react under heating module)
$ $	Characterization	$ABS - CD R_1 \times A$ (Take ABS-CD measurement under heating module)

Revised S2. Communication test

Higher data rates and lower delays are eager for this cloud lab. The burden is sometimes a big cable for flexible applications. Antenna based mobile networks are proved efficient in robotic automation. 5G technology is regarded as the revolution power for the future robotic industry. Here, we tested the performance of three different generations of networks collecting users and cloud lab (Shown in Figure S3 and Table S3). Under 5G (provided by China Unicom &

Ericsson), 405Mbps data rate enables real-time monitoring of 90-120fps video flow with high resolution (1920*1080) captured by an industrial camera. 19ms Ping also significantly enhanced the preciseness of remote control of robot cells.

Figure S3. Communication scheme of the system

Table S3. Summary of tested Ping value and bit rate through different mobile networks

	WLAN	4G	5G
Ping	39 ms	75ms	19ms
Transmission rate	140Mbps	88Mbps	405 Mbps

Comment 1-2: The authors explore a 2D search space (concentration vs temperature) using the SNOBFIT algorithm and 250 experiment loops. Given that this undoubtedly translates to a great number of experiments overall, and considering the rather simple search space, I wonder if the authors' algorithm is any better than a random search or even a human-driven (manual) search. These experiments should be performed for comparison.

Reply 1-2: We agree with the reviewer's suggestion that more details of SNOBFIT should be clearly stated. Actually, this algorithm⁸ was widely utilized in chemical reaction searching and optimization^{9, 10, 11}. In short, SNOBFIT combines the concepts of random search in the global region and gradient descent in the local region. Shown in revised Fig 2, sampling at the initial state of reaction searching is in high randomness. With about 50 trials, a local optimal region was firstly discovered at a

temperature around 110 °C. Then, after 30 more trails, another local region with higher CD intensity was found, precise gradient descent was applied here (closely located orange points) and achieved the CD strength reached 1.0 mdeg. So far, the searching and optimization progress took about 120 loops. Further, a global random search kept exploring the 2-D parameter space (sparsely distributed dark orange points on the map) and finally reached the pre-set max order (250 loops). The global optimal condition was proved to be the local region founded around 120 loops. The reviewer worried about that SNOBFIT took too many trails (250) here and didn't show 'intelligence' when compared with a random or human-driven search. Actually, as described above, the random search may help us to find some CD signal from synthesized IPNCs, but how to get enhanced optical activity still needs local optimization strategies guided by gradient descent algorithms. Additionally, similar with the most optimization algorithms, the trails can be significantly reduced by increasing the minimal step [dT, dC]. In this work, by considering the cost-efficient experiment process, we set a precise minimal step [dT = 0.5 °C, dC = 0.01 mmol/ml] to avoid miss of extrema. It took about 4 hours for completely ran the 120 loops discovery of the ever reported highest CD intensity from IPNCs, which proved the power of AI endorsed automation in chemical lab. Of course, the manual knowledge base of scientists is also a key parameter in experiment setup, boundary definition, algorithms design and theoretical analysis. The rational utilization of lab automation, the combination of AI algorithms and physical theories can truly speed up the materials discovery process.

Revised Fig 2. Results of autonomously discovered optical active IPNCs driven by MAOSIC. (a) SNOBFIT codes guided the chemical machines to search the two-dimensional parameter plane for a higher CD signal. (b)-(i) TEM images and CD spectrum of chiral IPNCs samples with various synthesis conditions.

Corresponding revision in the main text has been highlighted in yellow.

Revision in supporting information are listed below:

Revised S4. Algorithms, pseudocode and compile process

The parameters for the SNOBFIT based reinforcement learning (RF) in this work is of the form like $(\mathcal{S}, \mathcal{A}, [\mathcal{P}_{as}], \mathcal{R})$. Here \mathcal{S} denotes the set of state s , which is the set of all reaction

parameters and an experimental result that has been experimented. State at timestamp i is denoted by $s_i = [a_0, r_0, \dots, a_i, r_i]$; \mathcal{A} denotes the set of action a_i . In the context of reaction optimization. \mathcal{A} is the combination of reaction parameters such as reaction time and reactant concentration. Action at timestamp i is denoted by $a_i = [a_i^1, \dots, a_i^j, \dots, a_i^n]$; $[\mathcal{P}_{as}]$ denotes the state transition probability, here p_{as} decides what action a to make under the experimental condition s . This action decision policy is denoted by π ; \mathcal{R} denotes the reward function under state \mathcal{S} and action \mathcal{A} . In the environment of reaction optimization, the reward at timestamp i is denoted by $r_i = [r_i^1, \dots, r_i^j, \dots, r_i^n]$, the reward function denotes the mapping $r_i = R(a_i)$. At a specific experimental condition a_i is mapped to get the difference between the experimental and the target then get the reward r_i .

For a timestamp i with the previous state s_{i-1} , the policy function decides the action to make by $a_i = \pi(s_{i-1})$. Then, continue the experiment with a_i , and get the reward $r_i = R(a_i)$. For the MAOSIC, architecture, the state could be updated by $s_i = s_{i-1} \cup [a_i, r_i]$. Policy function π performs local optimizations around the best conditions while searching for unexplored regions to ensure global optimality. For local optimization, the action is decided by searching from the best condition with a full quadratic model. For example, an action could be decided by approximately minimizing the local quadratic model. $q(a) = r_{best} + g^T(a - a_{best}) + \frac{1}{2}(a - a_{best})^T G(a - a_{best})$, where g and G are the gradients and the symmetric matrix estimated, respectively.⁸

For the global optimization, the action is decided by searching simultaneously in several promising sub-regions. For example, the parameter space is partitioned into sub-boxes each contains a history state⁸. The actions are chosen by

$$a = \begin{cases} \frac{1}{2}(a_i + a_i), & \text{if } a_i - \underline{a}_i > \overline{a}_i - a_i \\ \frac{1}{2}(\overline{a}_i + a_i), & \text{otherwise} \end{cases}, \text{ where } [\underline{a}_i, \overline{a}_i] \text{ is one of the largest sub-boxes of}$$

the current partition with history action a_i inside. The actions selected will finally be rounded into the nearest integral multiple of the resolution vector Δa , which is decided by the physics condition of reaction parameters (here $\Delta a = (\Delta T_i, \Delta c) = (0.5^\circ\text{C}, 0.01\text{mmol/mL})$),

T_i and c are the heating temperature and reactant concentration respectively). We can make a specific reaction parameter Δa^j large enough, then the local optimization would have little influence on this dimension because of the round procedure. For the global search mentioned above, new actions would be selected in the most massive sub boxes.

Pseudocode for chiral CsPbBr₃ nanocrystals discovery

```
def HANDLE_REQUEST(request_id):
    target, reaction_id = FETCH_REQUEST_INFO(request_id)
    config = FETCH_REACTION_CONFIG(reaction_id)
    if not CHECK_RESOURCES(reaction_id):
        raise Exception
    reaction = GET_REACTION(config)
    optimizer = GET_OPTIMIZER(config)
    reaction.initialize()
    optimizer.initialize()
    LOAD_REACTION_HISTORY(optimizer, reaction_id)
    if OPTIMAL(optimizer.get_best()):
        return optimizer.get_best()
    n_call = 0
    while n_call < config.max_call:
        action = optimizer.choose_action()
        result = reaction.experiment(action)
        reward = reaction.analyzer.calc_reward(result,
target)
        STORE_RESULT(action, result)
        optimizer.update_env(action, reward)
        n_call = n_call + LENGTH(action)
        if OPTIMAL(optimizer.get_best()):
            return optimizer.get_best()
    return optimizer.get_best()
```

Notes:

FETCH_REQUEST_INFO:

load request information from the database.

FETCH_REACTION_CONFIG:

loads the configure file of the reaction from the database, which contains all the information about how to synthesis and the hyper-parameter of the optimizer.

CHECK_RESOURCES :

checks if the hardware works well and the reagent is enough.

The reaction in charge of communicating with the hardware control interface and controlling the experiment optimizer choose the reaction parameters (action) and learn from the analysis result (reward).

LOAD_REACTION_HISTORY:

load all experiments about this action and update the state of the optimizer.

STORE_RESULT:

store the experiment results in the database.

Compile in cloud server

An example of compiling the input experimental formula to machine instruction code is shown in Figure S9. This example shows how MAOSIC extracts information from input formula, and set up all experimental parameters in the configuration file for CsPbBr₃ nanocrystals synthesis. The configuration file was generated through the designed template. It decides all adjustable or constant parameters for optimizing the experiment.

Figure S9. Scheme of compiling process in MAOSIC. The color if icons and cods indicate different steps in the experiment. Firstly, the formula (on right) with vectors of module parameters (in each icon) was input to MAOSIC. Then, the instruction codes (on left) were generated through the compiling process.

Configuration file

```

{"type": "fluid reaction",
"name": "CsPbBr3 Nanocrstals",
"reagent": [{"formula": "Cs-Precursor", "ratio": 1},
{"formula": "Pb-Precursor", "ratio": 2},
{"formula": "ODE", "ratio": 1}],
"metric": ('CD', ['wavelength', 'peak']),
"param": {"temp": {"default": 100, "bound": [90, 180], "dx": 1, "var": 1},
"ODE": {"default": 1, "bound": [0.01, 100.0], "dx": 0, "var": 1},
"time": {"default": 20, "bound": [20, 20], "dx": 0, "var": 0}},
"init": ["PUMP_INIT"],
"experiment": ['FILL Cs-Precursor', 'FILL Pb-Precursor', 'FILL ODE', 'WAIT 15', 'HEAT $temp',
'WAIT 30', 'PUMP_PARALLEL $time 0.5', 'WAIT $time', 'PUMP ODE 0.5', 'WAIT 10', 'ABS-CD'],
"optimizer": "snobfit",
"max_expts": 250,
"nreq": 0}

```

Figure S10. An example configuration file for chiral CsPbBr₃ nanocrystals synthesis.

Figure S11. Workflow of the experiment driven by MAOSIC. **T** indicates the reaction temperature. **P1, P2, P3** refer to the injection rate of each syringe pumps, which controlled the reaction time and precursors concentration.

Comment 1-3: DFT calculations are performed but no detail is given about how these

were done. From the figure it seems that the authors used a cluster model to define the perovskite. If so, this requires justification.

Reply 1-3: For the DFT calculation, we use the Perdew–Burke–Ernzerhof (PBE)¹² exchange correlation functional in Quantum Espresso code¹³. A 60 Ry kinetic energy cut-off and Gamma K-point are applied for the ground state calculation. We build a Cs₅₇Pb₂₇Br₁₁₁ cluster with dislocation to mimic the chiral structures. As shown in Figure 3 (c), with the appearance of dislocation, the HOMO and LUMO distributions still lie on the Pb/Br sites, overlaps with the first absorption peak origins from Pb²⁺(6s)Br⁻(4p)→Pb²⁺(6p),¹⁴ indicating the dislocation didn't largely change the excitation characters. This description has been added to the Simulation method in the manuscript.

Reference:

1. Henson AB, Gromski PS, Cronin L. Designing Algorithms To Aid Discovery by Chemical Robots. *ACS Central Science* 2018, **4**(7): 793-804.
2. Ley SV, Fitzpatrick DE, Ingham RJ, Myers RM. Organic synthesis: march of the machines. *Angew Chem Int Ed* 2015, **54**.
3. Ley SV, Fitzpatrick DE, Myers RM, Battilocchio C, Ingham RJ. Machine- assisted organic synthesis. *Angew Chem Int Ed* 2015, **54**.

4. Tabor DP, Roch LM, Saikin SK, Kreisbeck C, Sheberla D, Montoya JH, *et al.*
Accelerating the discovery of materials for clean energy in the era of smart automation.
Nature Reviews Materials 2018, **3**(5): 5-20.

5. Roch LM, Häse F, Kreisbeck C, Tamayo-Mendoza T, Yunker LPE, Hein JE, *et al.*
ChemOS: Orchestrating autonomous experimentation. *Science Robotics* 2018, **3**(19):
eaat5559.

6. Fitzpatrick DE, Maujean T, Evans AC, Ley SV. Across-the-World Automated
Optimization and Continuous-Flow Synthesis of Pharmaceutical Agents Operating
Through a Cloud-Based Server. *Angewandte Chemie-International Edition* 2018,
57(46): 15128-15132.

7. Elvira KS, Casadevall i Solvas X, Wootton RC, deMello AJ. The past, present and
potential for microfluidic reactor technology in chemical synthesis. *Nature chemistry*
2013, **5**(11): 905-915.

8. Huyer W, Neumaier A. SNOBFIT -- Stable Noisy Optimization by Branch and Fit. *ACM
Trans Math Softw* 2008, **35**(2): 1-25.

9. Bédard A-C, Adamo A, Aroh KC, Russell MG, Bedermann AA, Torosian J, *et al.*
Reconfigurable system for automated optimization of diverse chemical reactions.
Science 2018, **361**(6408): 1220.

10. Holmes N, Akien GR, Blacker AJ, Woodward RL, Meadows RE, Bourne RA.
Self-optimisation of the final stage in the synthesis of EGFR kinase inhibitor AZD9291
using an automated flow reactor. *Reaction Chemistry & Engineering* 2016, **1**(4):
366-371.

11. Krishnadasan S, Brown RJC, deMello AJ, deMello JC. Intelligent routes to the
controlled synthesis of nanoparticles. *Lab on a chip* 2007, **7**(11): 1434-1441.

12. Perdew JP, Burke K, Ernzerhof M. Generalized Gradient Approximation Made Simple.
Physical Review Letters 1996, **77**(18): 3865-3868.

13. Giannozzi P, Andreussi O, Brumme T, Bunau O, Buongiorno Nardelli M, Calandra M,
et al. Advanced capabilities for materials modelling with Quantum ESPRESSO.
Journal of Physics: Condensed Matter 2017, **29**(46): 465901.

14. De Bastiani M, Dursun I, Zhang Y, Alshankiti BA, Miao X-H, Yin J, *et al.* Inside Perovskites: Quantum Luminescence from Bulk Cs₄PbBr₆ Single Crystals. *Chemistry of Materials* 2017, **29**(17): 7108-7113.
15. Mukhina MV, Maslov VG, Baranov AV, Fedorov AV, Orlova AO, Purcell-Milton F, *et al.* Intrinsic Chirality of CdSe/ZnS Quantum Dots and Quantum Rods. *Nano Letters* 2015, **15**(5): 2844-2851.
16. Müller G, Métois JJ, Rudolph P. *Crystal Growth - From Fundamentals to Technology*. Elsevier Science, 2004.
17. Nagaev E. Light-induced phase transitions. *Zh Eksp Teor Fiz* 1981, **80**: 2346-2355.
18. Li J, Bo B, Gao X. The degradation of structure and luminescence in CsPbBr₃ perovskite nanocrystals under UV light illumination. *AIP Conference Proceedings* 2018, **2036**(1): 030018.
19. Liu J, Song K, Shin Y, Liu X, Chen J, Yao KX, *et al.* Light-Induced Self-Assembly of Cubic CsPbBr₃ Perovskite Nanocrystals into Nanowires. *Chemistry of Materials* 2019, **31**(17): 6642-6649.

20. JÓNSSON H, MILLS G, JACOBSEN KW. Nudged elastic band method for finding minimum energy paths of transitions. *Classical and Quantum Dynamics in Condensed Phase Simulations*, pp 385-404.
21. Kirschner MS, Diroll BT, Guo P, Harvey SM, Helweh W, Flanders NC, *et al.* Photoinduced, reversible phase transitions in all-inorganic perovskite nanocrystals. *Nature Communications* 2019, **10**(1): 504.
22. Huang Y, Wang L, Ma Z, Wang F. Pressure-Induced Band Structure Evolution of Halide Perovskites: A First-Principles Atomic and Electronic Structure Study. *The Journal of Physical Chemistry C* 2019, **123**(1): 739-745.
23. Bekenstein Y, Koscher BA, Eaton SW, Yang P, Alivisatos AP. Highly Luminescent Colloidal Nanoplates of Perovskite Cesium Lead Halide and Their Oriented Assemblies. *Journal of the American Chemical Society* 2015, **137**(51): 16008-16011.
24. Bausch R, Schmitz R, Turski ŁA. Single-Particle Quantum States in a Crystal with Topological Defects. *Phys Rev Lett* 1998, **80**(11): 2257-2260.

25. Baimuratov AS, Rukhlenko ID, Noskov RE, Ginzburg P, Gun'ko YK, Baranov AV, *et al.*
Giant Optical Activity of Quantum Dots, Rods and Disks with Screw Dislocations.
Scientific Reports 2015, **5**(1): 14712.

26. Baimuratov AS, Pereziabova TP, Leonov MY, Zhu W, Baranov AV, Fedorov AV, *et al.*
Optically Active Semiconductor Nanosprings for Tunable Chiral Nanophotonics. *ACS*
Nano 2018, **12**(6): 6203-6209.

27. Berova N, Bari LD, Pescitelli G. Application of electronic circular dichroism in
configurational and conformational analysis of organic compounds. *Chemical Society*
reviews 2007, **36**(6): 914-931.

28. Hirth JP, Lothe J, Mura T. Theory of dislocations. *Journal of Applied Mechanics* 1983,
50: 476.

Reviewer #2 (Remarks to Authors):

The authors have written an article describing the development of a fully automated robotic setup using algorithms to produce high throughput research, testing a wide range of preparation conditions for CsPbBr₃ nanoparticles.

They have built a setup from scratch, incorporating a range of preparation and characterisation techniques to form a fully automated preparation and characterisation workflow which can be controlled remotely. The setup is able to control the reaction temperature and precursor concentration with very high accuracy, and a feedback algorithm allows the system to iteratively test out different parameter combinations in order to study a range of properties.

In this case, the authors record the absorption spectrum and the circular dichroism (CD) and find a preparation condition with a high degree of circular dichroism at around 130 °C and 0.8 mmol/mL precursor solution. They then characterise the nanoparticle produced at this condition further using high resolution TEM (HR TEM) and claim to observe the presence of screw dislocations in the nanoparticles, to which they attribute the circular dichroism observed. Finally, they induce changes in the polarisation direction of the nanoparticles by laser pulsing, and set up a quantum mechanical model describing the change in dichroism as a function of temperature.

The combination of robotics and iterative AI use for material synthesis and characterisation is very impressive, and the setup is an engineering feat and could form the basis of much future material development and characterisation. The ability to test hundreds of parameter combinations in a short period of time is very promising and has the potential to fundamentally change how these materials are studied.

Similarly, the ability to measure absorption and circular dichroism on the fly, and using the information to prepare the next solution, effectively honing in on an optimal recipe is

impressive, and has allowed to authors to synthesise a compound with a high degree of circular dichroism, a phenomenon which is not widely seen in the perovskites usually studied today.

The inorganic perovskite nanocrystals typically showing CD usually have chiral ligands or organic molecules associated with them, but the authors claim to synthesise fully inorganic nanoparticles with no ligands, which still show CD. They attribute this to the presence of screw dislocations, as theorised would be possible in inorganic materials, and shown in highly chiral nanowires in refs. 27 and 28, respectively. The authors claim to observe a same kind of screw dislocation-induced chirality.

The paper is overall quite well written, and only needs minor editing for typos and clarifications. The sections about the automated setup are in my view exceptional, although I do admit not to have a strong background in robotics and automation. I also do not have a strong background in quantum mechanics, and I hope another reviewer will be qualified to assess this work.

The sections related to the HR TEM results do, however, require some further analysis and/or explanations. The presence of CD is an interesting property, but there are a range of questions that must be addressed regarding these results and their interpretation before the paper can be considered for publication in Nature Communications.

Below are some general points, followed by text-specific points with references to the line number in the pdf version.

Comment 2-1: The evidence for screw dislocations in the nanowires is not clear at best, and if the dislocations were the origin of the chirality, their random distribution would intuitively cancel out any local chirality. The authors have drawn red dotted lines in Figure 3, which are supposed to represent screw dislocations, but the lines seem to follow the surface of the nanoparticle more than anything, as seen by the amorphous region (probably the carbon grid) seen next to the supposed dislocations. If the nanoparticles contained screw dislocations, they

would be expected to be found within the crystal structure itself, and not at the edges of it.

Reply 2-1: Thanks for your comments. From the view of statistics, the random distribution of screw dislocation does intuitively cancel out local chirality. But actually, in this experiment, a significant CD signal was collected. The driven force which induced difference between L- and D-IPNCs during synthesis still needs investigation in the future. This is similar to the famous problem in the field of life science that most living beings on earth are constructed by L-amino acids. The origin of this kind of chirality separation in life is also under debating.

As for the identification of the dislocations, we accept your suggestion that they should be found within the crystal structure. We referred related work about intrinsic chirality of CdSe/ZnS nanocrystals from Yurri Gunko¹⁵ and revised Figure 3. Corresponding correlations in manuscript are highlighted in yellow.

Revised Fig. 3 Structure identification of synthesized chiral CsPbBr₃ nanocrystal sample. (a) Cubic phase of CsPbBr₃ crystalline was determined from XRD pattern. (b) Chiral assembly of rod-like CsPbBr₃ nanoplates (local optimal sample) were founded under TEM Images. (c) Evidence of screw dislocation measured by high resolution transmission electron microscope (HRTEM) in global optimal sample. (110) and (200) phases were measured to confirm the cubic crystal structure. (d)-(e) Schematic model of CsPbBr₃ nanocrystal with screw-dislocation. The curve lattices along (200) phase are marked with red points corresponding to the TEM image, which indicates a possible screw dislocation in nanocrystal. The screw-line is highlighted with amplified Pb-Br atoms, perpendicular to Burgers vector. The shear interface is drawn in blue.

Comment 2-2: The authors apply pulsed laser on the nanoparticles to induce thermal vibrations in the nanoparticles, and observe a change in chirality as a result. It is not clear what the exact purpose of the pulsed laser is, other than thermal activation, in which case it should be stated which temperature the particles ended up at. The laser treated particles experience a change in chirality, which the authors explain with the quantum mechanical model, showing that chirality can change as a function of temperature, with certain temperatures corresponding to certain inversions. If the films were all treated at the same temperature, one would expect that only one chirality change would be possible (for example, at 75 °C D->L would be possible, but not L->D, according to Figure 4 (d)). If this is the case, only one type of inversion should be observed, but the authors observe inversions in both directions. This should be better explained in particular, and the chirality change could be better explained in general.

Reply 2-2: We chose 800nm femtosecond pulse laser (1000Hz) to enhance the thermal vibration of the dislocations in nanocrystals. The femtosecond laser source is widely applied in fast events investigation of like electron dynamics, and in some laser-assisted surgery with high precise and tiny trauma requirements. Similarly, we choose it to affect the crystal dynamics because it may cause less damage to the sample itself. Additionally, the dislocation formation belongs to the “non-equilibrium” process¹⁶, in which cases the light induction takes more advantage than the temperature does¹⁷. Actually, we have also tried direct heating or 300nm laser to treat CsPbBr₃ samples. Irreversible degradation and phase transition were found, agree with related reports¹⁸. No chirality inversion was observed and CD signal reduced significantly. On the other hand, the CD characterization was not in-situ tools here, so even the CD changed at high temperature region, the testing environment still contained at room temperature. The experiment of temperature-dependent chirality of chiral CsPbBr₃ nanocrystals will be carried in future research. Glovebox environment with no H₂O and oxygen should be applied to keep the sample stable, and the heating equipment within CD spectrometer will be set up for in-situ characterization.

The CD inversion direction is related to the chirality of origin samples. Here we just plot one situation in Fig 4d, which is L-D-L-D. So of course, if the original sample at room temperature was D, the temperature dependence should change into D-L-D-L.

Comment 2-3: A significant change in morphology is seen after the laser treatment, seen in Figure 4 (c). The authors state that the overall phase is the same after the treatment, namely the cubic CsPbBr₃ phase, with small amounts of Cs₄PbBr₆ present on the surface of the newly formed particles, as evidenced by XRD. The new morphology seems so fundamentally different to the original one that this claim is hard to understand. There are quite clearly at least two different types of nanoparticles present, but it is difficult to impossible to examine them in detail in the current micrograph. The smaller particles do not show the same fringes as they did in Figure 2 (a) before treatment, so it is possible that they have amorphised as a result of the laser treatment. The new phases after laser treatment must be properly accounted for and described. If, for instance, it is the Cs₄PbBr₆ compound that displays chirality, the findings do not pertain to perovskite CsPbBr₃ nanoparticles.

Reply 2-3: The photon-induced assembly of CsPbBr₃ nanocrystals is observed under TEM after laser irradiation. This phenomenon has been reported by Liu et al in this year¹⁹. 1.5 AM solar simulator was utilized to treat the CsPbBr₃ nanocrystals samples and finally obtained self-assembled CsPbBr₃ nanowires. They attribute this assembly to the light-induced ligand removal mechanism. Using atomic-resolution electron microscopy, they visualized the cubic-to-orthorhombic phase transition in NCs and the interface between coalesced NCs.

The dim of fringes in nanocrystals was related to the focus under TEM. The focus point was located on the assembled structures with larger thickness, so that the fringes of smaller ones are not clear. Furthermore, DFT calculation was added to give evidence of the mixed Cs₄PbBr₆ peaks in XRD pattern after assemble (added in Supporting information S7).

Figure S13. crystal structure for (a) Cs_4PbBr_6 and (b) cubic CsPbBr_3 . The brown and gray color represents the Br and Pb respectively. The Cs atoms are not shown for clarity.

Figure S13 shows the crystal structure and local chemical bond information for Cs_4PbBr_6 and cubic CsPbBr_3 respectively. The main difference between the two phases is that the PbBr_6 octahedrons share vertexes in CsPbBr_3 and the same PbBr_6 octahedrons are isolated in Cs_4PbBr_6 . For the phase transition from CsPbBr_3 to Cs_4PbBr_6 , some Pb-Br bonds have to be broken in the CsPbBr_3 crystal phase to form the isolated PbBr_6 octahedron, and more Br- ions are essential for the stoichiometry, thus the octahedrons in the edge of CsPbBr_3 structure are easier to form the PbBr_6 octahedron with the external Br- ions from environment. In order to confirm the energy scale for the Cs_4PbBr_6 phase to emerge, we apply Nudge Elastic Band (NEB) ²⁰calculations with eight transition state images for the energy barrier investigations. As shown in Figure S14, here we use a $\text{Cs}_{21}\text{Pb}_8\text{Br}_{37}$ nanocluster to mimic the locale phase transitions. To form an isolated PbBr_6 octahedron, it needs one additional Pb-Br bond, which requires an additional Br- ion from the external environment. As indicated by the red circle in structure '1' and '2', once the Br- ion approaches the Pb atoms, a newly-formed Pb-Br bond comes into being, as indicated by the red bond in structure '3', and a Pb-Br bond previous shared by the nearby octahedrons is broken, as shown by the blue color in the same structure. Eventually the Br- ion from environment comes into bonding and the structure '4'

becomes more Cs_4PbBr_6 -like. The energy barrier for this processing is about 0.42 eV. We think the occurrence of Cs_4PbBr_6 phase is due to the event described in Figure S14, where the energy barrier is overcoming by the irradiation or consequential thermal fluctuation.

Figure S14. NEB calculations for the phase transition from CsPbBr_3 to Cs_4PbBr_6 . For simplicity, A $\text{Cs}_{21}\text{Pb}_8\text{Br}_{37}$ nanocluster is used for the simulation. The structure '1' and '4' represents the initial (CsPbBr_3) and final (Cs_4PbBr_6) state, '2' and '3' represent the middle stage structures. The red circle in structure '1' and '2' represent the Br^- ion from the environment. The red and blue bonds in structure '3' indicate the newly-formed bond from Pb and external Br^- ion and tentatively-broken Pb-Br bond. The energy barrier is about 0.42 eV.

The chirality inversion does not attribute to the small amount of Cs_4PbBr_6 phase, as the optical absorption of Cs_4PbBr_6 nanocrystals is in the UV region ($>100\text{nm}$ blue-shift from 450nm).

Comment 2-4: There is some important experimental information missing, such as the microscope conditions used (including microscope type, voltage, dose rate, etc.) and the TEM sample preparation methods. How were the nanocrystals deposited on the TEM grids?

Reply 2-4: Sorry for missing such important information. Sampling and characterization tools are added in Supporting Information S4:

TEM, FEI Tecnai G2 F30, High resolution TEM (HRTEM, FEI Tecnai G2 F30), 300kV, 120uA.

X-Ray Diffraction (XRD, Bruker Advance D8 Ew, Germany). 300KV

The purification and TEM sampling process of as-synthesized IPNCs: Mix with ethyl acetate with 1:1 ratio and centrifuge at 8000rpm for 4min. Take the deposited sample and dissolved in 5ml n-hexane. Centrifuge for 2min at 4000rpm. Deposit the solution on the ultrathin carbon grid for natural volatilization.

Comment 2-5: 64: It would be useful if MAOSIC is defined in the main text as well as/instead of in the abstract.

Reply 2-5: Thanks for your comment. The correlation has been made. MAOSIC has been defined in the main text as well as in the abstract.

In abstract: Driven by the Materials Acceleration Operation System In Cloud (MAOSIC), on-demand experimental design from remote users was authentically carried out in this cloud lab.

In the main text: Driven by Materials Acceleration Operation System In Cloud (MAOSIC), this cloud lab has provided an authentic synthetic platform for cross-world researchers.

Comment 2-6: 83: 'sent' should probably be 'send'.

Reply 2-6: Thanks for your comment. The correlation has been made.

Comment 2-7: Figure 1: What happened to the spent materials?

Reply 2-7: The spent materials in cuvette will be handled out from CD spectrometer by the robot, and then be poured out into a waste container.

Comment 2-8: Figure 2: Are (b,c) measured at room temperature? Which synthesis temperature(s) was/were used? Units on the concentration legends would be useful.

Reply: The concentration-dependent absorption and CD measurement (Figure 2b and c) were taken at room temperature. The synthesis temperature is around 130°C (data was collected from the original synthesis process (in 250 loops) during the recipe optimization guided by SNOBFIT algorithms, to be exact, the temperature region within 130 ± 5). Also, we revised Figure 2 with additional TEM images.

Comment 2-9: 130-132: Which is the first excitonic peak? Why only the first one? 0.8 mmol/mL has both a positive and a negative peak, which one was used? If the positive peak, how can it be compared to the other, negative ones. If the negative one, what does the positive mean in this case?

Reply 2-9: The words 'first excitonic peak' has been changed into 'first exciton absorption peak', which refers to the photon energy absorbed when the ground state electron transitions to the first excited state, and is shown as the first exciton absorption peak in the absorption spectrum. The collected positive and a negative peak was caused by the cotton effect (will be discussed in **Reply 2-11**). As described in the manuscript, to discover large optical activity in IPNCs, the automatic data analysis just recognizes the max intensity, which is the absolute

value of CD as a result to evaluate the experimental condition. So the chiral direction is not considered in the automatic experiments, but be discussed in the theoretical analysis section.

Comment 2-10: 142: 'crystallinities' should probably be 'crystal quality'.

Reply 2-10: Thanks for your comment. The correlation has been made.

Comment 2-11: 148: What is a 'cotton effect'? How does it show?

Reply 2-11: The Cotton effect is the characteristic change in optical rotatory dispersion and/or circular dichroism in the vicinity of an absorption band of a substance. In a wavelength region where the light is absorbed, the absolute magnitude of the optical rotation at first varies rapidly with wavelength, crosses zero at absorption maxima and then again varies rapidly with wavelength but in the opposite direction. It is positive if the optical rotation first increases as the wavelength decreases while negative if the rotation first decreases.

Comment 2-12: 148-150: This sentence is not clear and could use some rewording and a reference.

Reply 2-12: Thanks for your comment. The sentence has been changed into:

Moreover, the CD spectrum presents a cotton effect at a concentration close to 0.8 mmol/mL.

These CD signals were affected by the splitting of the excited states of two degenerate exciton-coupled chromophores, which is often observed in organic assemblies.

Comment 2-13: 156: 'HUMO' should probably be 'HOMO'.

Reply 2-13: Thanks for your comment. The correlation has been made.

Comment 2-14: Figure 3: There seems to be a shoulder to the left of the (100) peak, what does that correspond to?

(c) The simulation seems to be done on the orthorhombic phase since the Pb-Br bonds are bending? Why is this? Does this affect the simulation results?

Is the simulation done on four unit cells only? It seems odd that the LUMO/HOMO would be localized in each end of the box, and not distributed symmetrically. Why is this?

(d) What are the experimental conditions these images were taken under? What dose rate and total dose was used? How were the crystals transferred to the TEM grids? The red dots drawn on the figure seem to follow the surface/edge of individual nanorods and not screw dislocations within the crystals. There seem to be some kind of defects/dislocations just above the scale bar on the left and between the two red dotted lines on the right, but it is difficult to verify if these are screw dislocations from the given image. Is there other evidence for the presence of screw dislocations?

If screw dislocations are present in both directions, intuitively no overall chirality expected.

Reply 2-14: The small shoulder to the left of (100) peak may attribute to the orthorhombic phase during low sample preparation and testing²¹.

It is difficult for DFT to handle the chiral screw dislocation enclosed calculation due to the huge structure size. We build a $\text{Cs}_{57}\text{Pb}_{27}\text{Br}_{111}$ cluster with dislocation to mimic the chiral structures, this cluster is not symmetrical, and it is close to the orthorhombic phase but different from the symmetrical cubic one, however, for both of the two phases, the HOMO/LUMO distribution is of the same atomic sites²², from $\text{Pb}^{2+}(6s)\text{Br}^-(4p)$ to $\text{Pb}^{2+}(6p)$ ¹⁴.

The key point for this DFT calculation is to identify that with the screw dislocation, the HOMO/LUMO still distributes all over the whole structures, considering the fact the dislocation area only occupies limited part in the whole structure, it is obvious that the dislocation didn't largely change the excitation characters when viewed in energy space. This conclusion can be confirmed by the absorption peak in Figure 4, the absorption peak and CD peak are both around 450 nm. Similar to many previous literatures¹⁵, the screw dislocation takes more effect in chiral (L or D) selection but affects little in excitation dependent peak position.

This sample was synthesized at 130 °C and 0.8 mmol/mL precursor solution. The TEM characterization detail has been added in supporting information S4.

The revised Figure 3 updates the probable dislocation within the nanocrystals and been discussed in **Reply 2-1**.

The purification and TEM sampling process of as-synthesized IPNCs: Mix with ethyl acetate with 1:1 ratio and centrifuge at 8000rpm for 4min. Take the deposited sample and dissolved in 5ml n-hexane. Centrifuge for 2min at 4000rpm. Deposit the solution on the ultrathin carbon grid for natural volatilization. The type and detail parameters of this HRTEM is FEI Tecnai G2 F30, with voltage 300kV, dose rate 120uA.

From the view of statistics, the random distribution of screw dislocation does intuitively cancel out local chirality. But actually, in this experiment, a significant CD signal was collected. The driven force which induced difference between L- and D-IPNCs during synthesis still needs investigation in the future. This is similar with the famous problem in the field of life science that most living beings on earth are constructed by L-amino acids. The origin of this kind of chirality separation in life is also under debating.

Comment 2-15: 166-169: There is no justification in the text or reference explaining why the supposed locations of the HOMO and LUMO would somehow connect screw dislocations with optical activity in the nanocrystal.

Reply 2-15: From Figure 4, we can see the absorption peak and CD peak are both around 450 nm, which is almost the same value for the CsPbBr₃ crystals without dislocation²³. This conclusion indicates that when viewed in the energy space, the excitation event in the structure with dislocation may differ little with the structure without dislocation. The HOMO/LUMO plot partially supports the conclusion that with the screw dislocation, the HOMO/LUMO still distributes on the Pb and Br sites, very similar to the case of perfect

crystals¹⁴. While in geometry space, we can see the key role of the screw dislocation on the CD signal lies in the fact that the screw shape introduces the chiral symmetry for the electron scattering events more dependent on geometry, as discussed in many literatures^{24, 25, 26}.

Comment 2-16: 170-171: Where are the vacancies and self-assembled screw dislocations shown in the document?

Reply 2-16: Here has been correlated to avoid misunderstanding.

Comment 2-17: 171-172: Is there any evidence or reference for this claim?

Reply 2-16: Has been claimed in **Reply 2-15**.

Comment 2-18: 174: Where is this claim illustrated or supported?

Reply 2-18: This claim is just a conclusion of the above text that we have synthesized the chiral IPNCs with optical activity. We have revised this sentence: Since then, MAOSIC has achieved chirality in IPNCs, however, the optical rotation direction of synthesized samples was of high randomness.

Comment 2-19: 174-175: Does this mean that it was either left or right polarised within an entire sample, but it was difficult to control exactly which one of the polarisations came out, or that the polarisation overall was random within a sample?

Reply 2-19: Exactly what you mean, the polarization direction of samples cannot be predicted during synthesis. So, a chirality separating/adjusting method is needed here, which is the femtosecond laser irradiation utilized in this section.

Comment 2-20: 178-180: The previous sentence stated that no overall chirality was possible to observe. How can the pulse inverse it if it cannot be distinguished initially? Why was laser used and not just thermal heating?

Reply 2-20: The previous sentence does not mean no overall chirality can be observed. CD signal exactly exists, but the polarization direction of samples cannot be controlled before synthesis. As discussed in **Reply 2-2**, the femtosecond laser source is widely applied in fast events investigation of like electron dynamics, and in some laser-assisted surgery with high precise and tiny trauma requirements. Similarly, we choose it to affect the crystal dynamics because it may cause less damage on the sample itself. Actually, we have also tried direct heating or UV irradiation to treat CsPbBr₃ samples. Irreversible degradation and phase transition were found, agree with related reports¹⁸. No chirality inversion was observed and CD signal reduced significantly. On the other hand, the CD characterization was not in-situ tools here, so even the CD changed at high-temperature region, the testing environment still contained at room temperature. The experiment of temperature-dependent chirality of chiral CsPbBr₃ nanocrystals will be carried in future research. Glovebox environment with no H₂O and oxygen should be applied to keep the sample stable, and the heating equipment within CD spectrometer will be set up for in-situ characterization.

Comment 2-21: 181: It would probably be good to define what 'L' and 'D' refer to. It is implied due to the chirality discussion, but not explicit.

Reply 2-21: Thanks for your comment. The correlation has been made.

Comment 2-22: 181: Which figure does 'The insert picture' refer to? Figure 4 (a) and (b)? It should be stated explicitly.

Reply 2-22: The insert picture indicates the absorption spectrum in both Figure 4a and b. We revised this sentence as follow to avoid misunderstanding.

The peak wavelength and intensity of optical absorption vary less than 1% (the insert pictures in both Figure 4a and b), and the CD strength after irradiation was similar to the original intensity, which means the nanostructure of nanoplates kept unchanged during the laser treatment.

Comment 2-23: Figure 4: (a-b) The inversions happen at different wavelengths, why would this be? Are these different concentrations? If so, which? Why do both the blue and green data cross the 0.0 mdeg in (b)? What temperature was achieved during the laser treatment? If the same temperature was achieved, there should only be one chirality inversion observed.

(c) The nanoparticles here do not look anything like the ones seen in Figure 3 (a,d). Where is the evidence that the edges are Cs₄PbBr₆ and everything else is CsPbBr₃? The XRD pattern changes more than just the appearance of a few small peaks. The big peak around 30 degrees seems to be two peaks on top of each other and a very sharp, small peak around 20 degrees is present. Are these consistent with Cs₄PbBr₆? The ordering in the larger rods seems visible at the lower magnification, but not in the smaller rods. Are these still crystalline, or have they amorphised?

The y-axes of (a,b) are 'CD' but in (d) it is 'CD strength'. Is this on purpose? If so, what is the difference?

Reply 2-24: The difference of peak wavelength was related to different synthesis conditions. The sample in Figure 4a was under 110□ with 0.4mmol/ml concentration. The sample in Figure 4b was under 130□ with 0.8mmol/ml concentration.

The chirality inversion within certain temperature region was related to its original chirality. As discussed in **Reply 2-2**, Figure 4i (revised Figure 4d) just plot one situation, which is L-D-L-D. So of course, if the original sample at room temperature was D, the

temperature dependence should change into D-L-D-L. So two kinds of chirality inversion can be observed at the same temperature condition.

The main part of assembled nanostructure was proved to be CsPbBr₃ phase but not amorphous structure, as discussed in **Reply 2-3**. The two peaks around 30 degrees after irradiation agrees well with the XRD pattern of the orthorhombic CsPbBr₃ phase, which has been reported¹⁹. It's difficult to prove the Cs₄PbBr₆ phase is on the edge of the assembly. The evidence of Cs₄PbBr₆ phase is the (012) peak at 12 degrees in the XRD pattern. DFT calculation was added to give evidence of the mixed Cs₄PbBr₆ peaks in the XRD pattern after assemble. This part is discussed detailed in **Reply 2-3**.

The words 'CD' and 'CD strength' are in the same meaning. We unify them into 'CD' to avoid misunderstanding.

Comment 2-24: 277: The units given in the figure are mdeg. If they are arbitrary, they must be normalised relative to some reference value.

Reply 2-24: The units mdeg are not arbitrary.²⁷ The interactions of any chiral non-racemic sample with left and right circularly polarized light beams, i.e., two chiral physical entities one being the mirror image of the other, are of a diastereomeric type. Accordingly, the circular dichroism is defined as the difference

$$CD = A^l - A^r \quad (1)$$

where A^l and A^r are the absorptions of left and right circularly polarized light, respectively. For historical reasons, the output of CD instruments is usually measured as ellipticity θ (in mdeg), related to CD through θ (mdeg) = 33000 CD. In analogy to the Lambert and Beer law, one can define a molar quantity

$$\Delta\varepsilon = \varepsilon^l - \varepsilon^r = \frac{CD}{c \cdot b} \quad (2)$$

which is independent of the concentration c , expressed in mol·L⁻¹, and of the pathlength b , expressed in cm. The definition of eq (1) immediately tells us that CD can be measured only

in correspondence to absorption bands; a dichroic peak is also called a Cotton effect, on account of the discoverer of the phenomenon. It is worth observing that CD is a signed quantity, because ϵ^l may be smaller or larger than ϵ^r (and consequently A^l and A^r); it is easy to show that for each absorption band, the CD of two enantiomers is always exactly the opposite.

Comment 2-25: 278-279: Is this preparation temperature or experimental temperature?

Reply 2-25: The temperature means the simulation temperature in CD measurement, not the reaction temperature during synthesis.

Reference:

1. Mukhina MV, Maslov VG, Baranov AV, Fedorov AV, Orlova AO, Purcell-Milton F, *et al.* Intrinsic Chirality of CdSe/ZnS Quantum Dots and Quantum Rods. *Nano Letters* 2015, **15**(5): 2844-2851.
2. Müller G, Métois JJ, Rudolph P. *Crystal Growth - From Fundamentals to Technology*. Elsevier Science, 2004.
3. Nagaev E. Light-induced phase transitions. *Zh Eksp Teor Fiz* 1981, **80**: 2346-2355.
4. Li J, Bo B, Gao X. The degradation of structure and luminescence in CsPbBr₃ perovskite nanocrystals under UV light illumination. *AIP Conference Proceedings* 2018, **2036**(1): 030018.
5. Liu J, Song K, Shin Y, Liu X, Chen J, Yao KX, *et al.* Light-Induced Self-Assembly of Cubic CsPbBr₃ Perovskite Nanocrystals into Nanowires. *Chemistry of Materials* 2019, **31**(17): 6642-6649.

6. JÓNSSON H, MILLS G, JACOBSEN KW. Nudged elastic band method for finding minimum energy paths of transitions. *Classical and Quantum Dynamics in Condensed Phase Simulations*, pp 385-404.

7. Kirschner MS, Diroll BT, Guo P, Harvey SM, Helweh W, Flanders NC, *et al.* Photoinduced, reversible phase transitions in all-inorganic perovskite nanocrystals. *Nature Communications* 2019, **10**(1): 504.

8. Huang Y, Wang L, Ma Z, Wang F. Pressure-Induced Band Structure Evolution of Halide Perovskites: A First-Principles Atomic and Electronic Structure Study. *The Journal of Physical Chemistry C* 2019, **123**(1): 739-745.

9. De Bastiani M, Dursun I, Zhang Y, Alshankiti BA, Miao X-H, Yin J, *et al.* Inside Perovskites: Quantum Luminescence from Bulk Cs₄PbBr₆ Single Crystals. *Chemistry of Materials* 2017, **29**(17): 7108-7113.

10. Bekenstein Y, Koscher BA, Eaton SW, Yang P, Alivisatos AP. Highly Luminescent Colloidal Nanoplates of Perovskite Cesium Lead Halide and Their Oriented Assemblies. *Journal of the American Chemical Society* 2015, **137**(51): 16008-16011.

11. Bausch R, Schmitz R, Turski ŁA. Single-Particle Quantum States in a Crystal with Topological Defects. *Phys Rev Lett* 1998, **80**(11): 2257-2260.

12. Baimuratov AS, Rukhlenko ID, Noskov RE, Ginzburg P, Gun'ko YK, Baranov AV, *et al.* Giant Optical Activity of Quantum Dots, Rods and Disks with Screw Dislocations. *Scientific Reports* 2015, **5**(1): 14712.

13. Baimuratov AS, Pereziabova TP, Leonov MY, Zhu W, Baranov AV, Fedorov AV, *et al.* Optically Active Semiconductor Nanosprings for Tunable Chiral Nanophotonics. *ACS Nano* 2018, **12**(6): 6203-6209.

14. Berova N, Bari LD, Pescitelli G. Application of electronic circular dichroism in configurational and conformational analysis of organic compounds. *Chemical Society reviews* 2007, **36**(6): 914-931.

Reviewer #3 (Remarks to the Author)

In general the grammar needs improvement, in some cases the grammar impedes understanding of the paper. The authors should also be careful to define all the acronyms used in the manuscript.

Comment 3-1: There are a few claims about the material produced that are not well-supported. First, based off the TEM images in Figure 3, the authors claim platelets are produced, but give no evidence to differentiate the material from nanorods. Also, in their mathematical model a nanorod is assumed. The presence of platelets is seen in Figure 4, but here the image is of an irradiated sample, showing signs of degradation. So my overall question after reading the first section is whether the synthesis has produced platelets or rods?

Reply 3-1: Thanks for your comments. We applied the cylindrical coordinates for the theoretic derivation, it covers both the platelets and rods shapes model. We have corrected the typo and uniformed the narration with “nanoplates”. Additional Tem images were added in revised Figure 2 and Figure 4 to give evidence of nanoplates.

Comment 3-2: After irradiation, there are platelets in the TEM image and peaks from Cs₄PbBr₆ (lines 186-188) in the XRD pattern. It seems likely that the irradiation is doing a lot more than switching chirality. How does the degradation of the nanocrystal material (whether it be platelets or rods) impact your measurement of chirality or the analysis of the model?

Reply 3-2: For the degradation, what we observed was the aggregation of small rods into the larger ones, shown in Figure S16.

Figure S16. TEM images of aggregation among CsPbBr₃ nanocrystals.

In revised manuscript, we investigated two kinds of chirality inversion model including the inversion of chiral assembly pattern and the inversed screw-dislocation. Evidence is in revised Figure 4.

Revised Fig. 4 Investigation of the laser-induced chirality inversion of IPNCs. Laser-induced chirality inversion was achieved both in (a) local optimal sample (from L to R) and (b) global optimal sample (from R to L). An inversion of orientation was observed comparing the TEM image of (c) original local optimal sample and (d) after laser heating. The evidence of dislocation inversion is also captured in HRTEM image of (e) global optimal sample (f) after irradiation. (g)-(h) Aggregation took place after laser treatment. XRD patterns were inserted to compare. (i) Result of numerical simulation of the temperature-dependent chirality. In the simulation, it was assumed that β reverses first and converges to zero, and $\omega_{harmonic}$ increases monotonically with temperature. It could be seen that the chirality reversed twice finally converges to zero.

We think the aggregation or phase transition is not the key issue of chirality inversion. According to our model below (added in Revised Supporting information S6), the screw dislocations are more likely to occur in materials whose thickness is smaller. It agrees well with our experimental data, i.e., after aggregation, the CD signal becomes a little bit weak.

Morphology and temperature dependence of the screw dislocations

Besides the chiral dichroism of the nanomaterials induced by the screw dislocations, the possibility of forming such a dislocation with respect to different morphologies and temperature also requires attention. It is hereby analyzed by comparing the energy resulted from the screw dislocation and the thermal fluctuation of the temperature, by adopting the most common expression $E_{thermal} = \frac{3}{2}nk_B T$.

As for the energy induced by the screw dislocation, it can be calculated from the distortion of the geometry of the nanomaterials and the stress induced. The energy has already been defined as the energy density per unit length as

$$e_{stress} = \frac{\mu b^2}{4\pi} \left(\ln \left(\frac{r_{outer}}{r_0} \right) - 1 \right),$$

where μ is the shear modulus, b is the magnitude of the burgers vector, r_{outer} is the outer most range of the dislocation on the radial direction of the cylinder (not the radius of the cylinder), and r_0 (r_0 is taken to be approximately b) and r_{outer} gives the bounded region of the screw dislocation²⁸. So, this energy is linear with the length of the vertical direction, where the screw dislocation occurs. If it is assumed that the screw dislocation occurs only for a finite length l , the energy would be $e_{stress}l$.

As for the thermal fluctuation,

$$E_{thermal} = \frac{3}{2}nk_B T = \frac{3}{2} \frac{\rho\pi R^2 L}{M} k_B T.$$

By comparing the magnitude of $E_{thermal}$ and $E_{dis} = e_{stress}l$ per site, and plot them with respect to both the radius R and the temperature T , it could be observed that: as the radius growing, i.e. the morphology approaches 2-D, the energy is at first very small because of the size, the energy of dislocation then increases but falls below the temperature effect afterwards, which means the defected morphology is more likely to occur at if the morphology is more similar to 2-D; as for the temperature, the energy of dislocation also falls below the thermal energy after reaching some threshold, indicating that the defects are more likely to occur at higher temperature.

Similar treatment under 300 nm wavelength femtosecond laser was also applied for both samples, no inversed CD signal was obtained. Transient absorption spectrum of the global optimal sample was collected. A 90% decrease of absorption at 5ps, and a 39% increase of average carries lifetime (Figure S15) indicates a exists of defects-induced intermediates states after 800nm laser heating.

Figure S15. Transient absorption spectrum (TAS) at 5ps and fitting results of average carries lifetime for global optimal sample (a) (b) after 800nm laser and (c) (d) 300nm laser irradiated. All characterization was taken under 800nm pump laser with 8mw power.

Comment 3-3: The claim that the Cs₄PbBr₆ is only on the “edge of aggregated plates” (line 187) is completely unfounded. Is there any evidence that just the edge of platelets would be phase changing, as opposed to whole nanocrystals under irradiation?

Reply 3-3:

Figure S13: crystal structure for (a) Cs_4PbBr_6 and (b) cubic CsPbBr_3 . The brown and gray color represents the Br and Pb respectively. The Cs atoms are not shown for clarity.

Figure S13 shows the crystal structure and local chemical bond information for Cs_4PbBr_6 and cubic CsPbBr_3 respectively. The main difference between the two phases is that the PbBr_6 octahedrons share vertexes in CsPbBr_3 and the same PbBr_6 octahedrons are isolated in Cs_4PbBr_6 . For the phase transition from CsPbBr_3 to Cs_4PbBr_6 , some Pb-Br bonds have to be broken in the CsPbBr_3 crystal phase to form the isolated PbBr_6 octahedron, and more Br- ions are essential for the stoichiometry, thus the octahedrons in the edge of CsPbBr_3 structure are easier to form the PbBr_6 octahedron with the external Br- ions from environment. In order to confirm the energy scale for the Cs_4PbBr_6 phase to emerge, we apply Nudge Elastic Band (NEB) ²⁰calculations with eight transition state images for the energy barrier investigations. As shown in Figure S14, here we use a $\text{Cs}_{21}\text{Pb}_8\text{Br}_{37}$ nanocluster to mimic the locale phase transitions. To form an isolated PbBr_6 octahedron, it needs one additional Pb-Br bond, which requires an additional Br- ion from the external environment. As indicated by the red circle in structure '1' and '2', once the Br- ion approaches the Pb atoms, a newly-formed Pb-Br bond comes into being, as indicated by the red bond in structure '3', and a Pb-Br bond previous shared by the nearby octahedrons is broken, as shown by the blue color in the same structure. Eventually, the Br- ion from the environment comes into bonding and the structure

'4' becomes more Cs_4PbBr_6 -like. The energy barrier for this processing is about 0.42 eV. We think the occurrence of Cs_4PbBr_6 phase is due to the event described in Figure S14, where the energy barrier is overcoming by the irradiation or consequential thermal fluctuation.

Figure S14. NEB calculations for the phase transition from CsPbBr_3 to Cs_4PbBr_6 . For simplicity, A $\text{Cs}_{21}\text{Pb}_8\text{Br}_{37}$ nanocluster is used for the simulation. The structure '1' and '4' represent the initial (CsPbBr_3) and final (Cs_4PbBr_6) state, '2' and '3' represent the middle stage structures. The red circle in structure '1' and '2' represent the Br^- ion from the environment. The red and blue bonds in structure '3' indicate the newly-formed bond from Pb and external Br^- ion and tentatively-broken Pb-Br bond. The energy barrier is about 0.42 eV.

This part has also been added into Supporting information S7

Comment 3-4: In Figure 4b, it looks like the "original" sample has a mixture of both D and L. Is there any control over which chirality is produced by the synthesis? Or do you produce a mixture in each batch? Can you separate them post-synthesis if you produce both? Please

comment on the composition of the as-synthesized sample.

Reply 3-4: To be exact, the original samples are a mixture of nanocrystals with both L and D chiral defects, as verified under HRTEM. The CD signal comes from the differences of these defects. The driven force which induced difference between L- and D-IPNCs during synthesis still needs investigation in the future. This is similar to the famous problem in the field of life science that most living beings on earth are constructed by L-amino acids. The origin of this kind of chirality separation in life is also under debating. However, laser irradiation provides us an approach to control the chirality of samples. As described in the manuscript, the chirality can be inversed after femtosecond laser treatment.

Comment 3-5: In the introduction (line 54) a phase transitions are listed. These are material-dependent and can't be quoted for ABX₃ perovskites in general. The authors should specify which material they looked up for the stated phase transition. On line 172, Figure 4e and 4f are cited, but do not exist.

Reply 3-5: Thanks for your comment. As the phase transition happened at 403K and 361K was data for CsPbBr₃ crystals, so 'ABX₃' should be changed into 'CsPbBr₃'. The sentence is revised as: In CsPbBr₃ perovskite system, cubic structure (Pm3m) is commonly formed at high temperatures. The phase transition to lower symmetry perovskite phases such as tetragonal (P4/mbm) and orthorhombic (Pnma) phase are happening at 403K and 361K, separately.

Figure 4 has been revised with correct legend.

Reference:

1. Hirth JP, Lothe J, Mura T. Theory of dislocations. *Journal of Applied Mechanics* 1983, **50**: 476.

2. JÓNSSON H, MILLS G, JACOBSEN KW. Nudged elastic band method for finding minimum energy paths of transitions. *Classical and Quantum Dynamics in Condensed Phase Simulations*, pp 385-404.

Reviewers' comments:

Reviewer #2 (Remarks to the Author):

Thank you to the authors for their revisions of the manuscript titled 'Autonomous Discovery of Optically-Active Chiral Inorganic Perovskite Nanocrystals Through Intelligent Cloud Lab'. I think the work is potentially very interesting, but there are a lot of highly specialised aspects to it which are not very clearly explained, and which might be difficult for the general readership of Nature Communication to relate to without further explanations. Since other reviewers seem to be more well-versed in the robotics side of the work, I will focus my comments on the crystallography and nanoparticle aspects of the work.

Overall, the paper has been improved on and now has a much better flow. There are still some typos and language issues which could easily be fixed by some thorough editing, which should probably be done before the paper is finally published for improved clarity and to reduce misunderstandings. For example, line 211 refers to 'supporting information S8', which is a section in the SI and not a figure. There is also a Figure S8, which is not related to line 211, so some explicit clarification would be good, for example writing 'Supporting Information Section S8'. I know this is a minor comment, but it helps remove confusion when a reader tries to correlate the main text and the SI.

More specifically to the results, I find it difficult to observe the screw dislocations from the micrographs included in the paper. In Figure 3 (c) there are curved red lines passing from one nanocrystal to another, marking a continuous transition from one crystal direction to another. While it does seem like the crystals are intergrown, the overlapping regions do not obviously show characteristics associated with screw dislocations. As is shown schematically in Figure 3 (e), a screw dislocation occurs when a plane in the crystal connects with a plane "above or below" what it should connect to within the crystal. This means that in a transmission microscope, both the "straight" crystal and the "bent" crystal should be visible on top of each other, creating a zig-zag pattern at the interface region, which then connects two straight crystal regions (see for example Figure 3 in Yang, H., Lozano, J., Pennycook, T. et al. Imaging screw dislocations at atomic resolution by aberration-corrected electron optical sectioning. *Nat Commun* 6, 7266 (2015) doi:10.1038/ncomms8266, <https://www.nature.com/articles/ncomms8266> or Figure 4.23 (b) in <http://www.engineeringenotes.com/metallurgy/metal-dislocations/edge-dislocation-in-crystals-metallurgy/43065>). In the micrograph in Figure 3 in Yang et al., two screw dislocations are observed, each connecting "straight" crystal regions with "bent" regions, which is very different from the single bend observed in the paper at hand. A single bend is more crystallographically akin to a twin growth, which has been observed in MAPbI₃, for example (Rothmann, M., Li, W., Zhu, Y. et al. Direct observation of intrinsic twin domains in tetragonal CH₃NH₃PbI₃. *Nat Commun* 8, 14547 (2017) doi:10.1038/ncomms14547, <https://www.nature.com/articles/ncomms14547>).

As such, it is very difficult to reliably identify the presence of screw dislocations in the global optimum samples based on the micrographs included in the paper at the moment. It is clear from the CD results that there is some kind of chirality induced in the sample and it probably has links to the shape of the crystals found in the sample, but at the moment, the evidence for the presence of screw dislocations is not conclusive. I cannot find any references in the paper explaining why the observed crystallography represents the presence of screw dislocations. References 37-39 discuss CD caused by screw dislocations, but these references are all based on computational studies, and none of the references contain any examples of micrographs like the ones presented in the paper. A "bent" crystal does not imply chirality since the orientation of the bend can be formed by rotation, which is not the case for a chiral system. A better justification of why this crystallography represents screw dislocations should be included, either in the form of less ambiguous micrographs or in the form of references determining this kind of growth to be related to screw dislocations, or another explanation should be given.

Furthermore, as is always the issue with TEM studies, the areas shown are very small and from the data at hand, it is difficult to evaluate if they are representative. For example, Figure 3 (b) and 4 (c,d) show a single group of chirally assembled nanocrystals each having a different chirality, but it is not possible to see other groups and assess whether they have the same or a different chirality. If the orientations are randomly distributed with a small majority in one direction or another, showing crystal assemblies with different orientations as evidence of orientation inversion is not evidence for a microscopic origin and can be misleading. Similarly, the lines drawn in Figure 4 (e,f) do not follow the same directions. In Figure 4 (e), the red line is parallel to the planes in the crystal, whereas in (f) it is parallel to the crystal in the top and perpendicular to the crystal in the bottom. If the line in (f) were perpendicular to the crystal planes in the bottom crystal, it would have the same curvature as the line in (e), and no inversion would be observed. This should be addressed.

Similarly, in Figure 4 (e,f), only one in the 20-30 nanocrystals seen in the lower magnification micrographs show the supposed screw "attachment". If this ratio is representative, is it enough to induce the changes in the CD? It is not obvious to me from the theory section how the CD depends on the number or ratio of chiral nanocrystals, but the strength of the CD seen in Figure 4 (b) is much larger than that predicted by the theory presented in Figure 4 (i), why is this? This should be commented on as well. A micrograph with a magnification somewhere between that of Figure 4 (e) and (g) could be included, and the nanocrystals with the additional growth could be highlighted along with their orientation to give a better understanding of the differences in orientation.

Similar to the initial version of the paper, it is difficult to see how the only change in microstructure after irradiation is the presence of a small amount of Cs₄PbBr₆ near the edges of the CsPbBr₃ nanocrystals (Figure 4 (g,h)). After irradiation, particles or rods of several hundred nm length are formed, and the relative ratios of the XRD peaks change significantly. It is clear that the overall intensity drops and a secondary peak appears near 30 degrees, with almost the same intensity as the original peak at that angle. The peak near 15 degrees also shifts to lower angles and there are additional significant peaks near 20, 26, 28, 33, and 37 degrees comparable in intensity to the one near 15 degrees. The NEB calculations shown in SI S7 do not obviously support the change in XRD pattern and microstructure observed in Figure 4 (g,h). All of this should be addressed.

When reading through the 'Eigenfunction and eigenvalues' section, I find it very difficult to follow which parts of it are general, already published knowledge, and which parts are new derivations used to explain the current system. The section is called 'Eigenfunction and eigenvalues' but these expressions are not used anywhere in the text other than in the title of the section. The whole derivation has very few explanations. For example, it is not clear to me from the text why ρ is an important parameter or why the Whittaker-M function blows up for the negative μ when $\rho = 0$, but not for the positive μ . Similarly, the Cotton effect is introduced without any explanation of what it is or what it means for the properties of the materials. It would be good to specify which parts of the section are new derivations, and which are based on literature, and include citations at the appropriate places, for example when introducing the Whittaker-M function, as well as other places.

The explanations in the Temperature Dependence section are better, but the concept of ω seems to appear just before Equation 9 and is not really explained. From the text, it is not clear to me how the Burger's vector, the vector that describes the screw dislocation, changes with temperature. Does the screw dislocation oscillate back and forth with changing temperature, or from the influence of an external electromagnetic field? It is claimed that the CD converges, but the graph in Figure 4 (i) stops at 250 C while having a negative slope and before any convergence is shown.

In SI S8 it is not clear what the green and blue lines represent in Figure S15 (a,c), and the scale bars in Figure S16 are too small to read. The 'w' in milli-watt should also be capitalised as 'mW'.

I will not go into details regarding the language editing, but I highly recommend a thorough editing for language and formatting.

Reviewer #3 (Remarks to the Author):

I attached a PDF file where I highlighted grammar/spelling mistakes that should be reviewed and fixed before submission. Also, in Figure S6b there is a part labelled as "Y-shaped value" should this be "Y-shaped valve"?

One confusion I have is about the two types of chirality discussed on pages 3, 9 and in Figure 4. The authors refer to "chiral assembled structures" but my understanding was that the chirality measurements are done in solution.

1. Is there any evidence that the assembled structures observed on the TEM grid correlate to an assembly in solution? When NCs are deposited on TEM grids they can assemble into superlattices, which are not correlated to an assembly in solution, but rather to the deposition technique.

2. On page 9, the authors refer to a "high concentration". The concentration controlled in the flow reactor is that of the precursors, but here they seem to be talking about the final concentration of NCs in solution. This point should be cleared up. Also, if the chirality depends on the concentration of product (As suggested in the 2nd to last sentence on page 9) then why can't you just adjust the concentration of NCs in solution after synthesis to increase the chiral signature detected? I think this is a different point to how the precursor concentration influences the reaction path and therefore chirality.

3. In Figure 4c and d: Was this sample irradiated on the TEM grid? If not, do you have reason to believe that the orientation of the nanorods on the grid is more influenced by the previous irradiation in solution as opposed to the dynamics of solvent drying during deposition?

On page 8, "the nanostructure of nanoplates kept unchanged during the laser treatment". I consider the screw dislocation to be part of the nanostructure of the NCs, so saying it is unchanged even though you are inverting the chirality of the screw dislocation is misleading.

For the temperature dependence of the chirality, is there a reason why the 800nm laser was chosen to induce the chirality change instead of just heating the sample to various temperatures? Based on your model in 4i, the temperature chosen should give you the control to form D or L based on the temperature, instead of just being able to switch from one to the other. If you have a mixed sample (containing both D and L), do all the NCs invert? Or do you see that they all tend to the same D or L chirality after laser exposure?

In Figure 4 caption it says that the chirality changes twice, but it looks like it inverts 3 times (L→D, D→L, and then L→D again) in 4i. Based on this model do you believe the chirality is determined by the synthesis temperature?

Reviewers' comments:

Reviewer #2 (Remarks to the Author):

Thank you to the authors for their revisions of the manuscript titled 'Autonomous Discovery of Optically-Active Chiral Inorganic Perovskite Nanocrystals Through Intelligent Cloud Lab'. I think the work is potentially very interesting, but there are a lot of highly specialised aspects to it which are not very clearly explained, and which might be difficult for the general readership of Nature Communication to relate to without further explanations. Since other reviewers seem to be more well-versed in the robotics side of the work, I will focus my comments on the crystallography and nanoparticle aspects of the work.

Overall, the paper has been improved on and now has a much better flow. There are still some typos and language issues which could easily be fixed by some thorough editing, which should probably be done before the paper is finally published for improved clarity and to reduce misunderstandings. For example, line 211 refers to 'supporting information S8', which is a section in the SI and not a figure. There is also a Figure S8, which is not related to line 211, so some explicit clarification would be good, for example writing 'Supporting Information Section S8'. I know this is a minor comment, but it helps remove confusion when a reader tries to correlate the main text and the SI.

Reply: Thanks for your suggestion. The relevant part has been revised. The description "supporting information Sx" has been changed into "Supporting Information Section Sx" and highlighted in the revised manuscript.

More specifically to the results, I find it difficult to observe the screw dislocations from the micrographs included in the paper. In Figure 3 (c) there are curved red lines passing from one nanocrystal to another, marking a continuous transition from one crystal direction to another. While it does seem like the crystals are intergrown, the overlapping regions do not obviously show characteristics associated with screw dislocations. As is shown schematically in Figure

3 (e), a screw dislocation occurs when a plane in the crystal connects with a plane “above or below” what it should connect to within the crystal. This means that in a transmission microscope, both the “straight” crystal and the “bent” crystal should be visible on top of each other, creating a zig-zag pattern at the interface region, which then connects two straight crystal regions (see for example Figure 3 in Yang, H., Lozano, J., Pennycook, T. et al. Imaging screw dislocations at atomic resolution by aberration-corrected electron optical sectioning. *Nat Commun* 6, 7266 (2015) doi:[10.1038/ncomms8266](https://doi.org/10.1038/ncomms8266), or Figure 4.23 (b) in <http://www.engineeringenotes.com/metallurgy/metal-dislocations/edge-dislocation-in-crystals-metallurgy/43065>). In the micrograph in Figure 3 in Yang et al., two screw dislocations are observed, each connecting “straight” crystal regions with “bent” regions, which is very different from the single bend observed in the paper at hand. A single bend is more crystallographically akin to a twin growth, which has been observed in MAPbI₃, for example (Rothmann, M., Li, W., Zhu, Y. et al. Direct observation of intrinsic twin domains in tetragonal CH₃NH₃PbI₃. *Nat Commun* 8, [14547](https://doi.org/10.1038/ncomms14547) (2017) doi:[10.1038/ncomms14547](https://doi.org/10.1038/ncomms14547), <https://www.nature.com/articles/ncomms14547>).

As such, it is very difficult to reliably identify the presence of screw dislocations in the global optimum samples based on the micrographs included in the paper at the moment. It is clear from the CD results that there is some kind of chirality induced in the sample and it probably has links to the shape of the crystals found in the sample, but at the moment, the evidence for the presence of screw dislocations is not conclusive. I cannot find any references in the paper explaining why the observed crystallography represents the presence of screw dislocations. References 37-39 discuss CD caused by screw dislocations, but these references are all based on computational studies, and none of the references contain any examples of micrographs like the ones presented in the paper. A “bent” crystal does not imply chirality since the orientation of the bend can be formed by rotation, which is not the case for a chiral system. A better justification of why this crystallography represents screw dislocations should be included, either in the form of less ambiguous micrographs or in the form of references determining this kind of growth to be related to screw dislocations, or another explanation should be given.

Reply: Thanks for this comment. As the reviewer described, to confirm the screw-dislocation directly from TEM images does not convincing. In the revised manuscript, we added more evidence that may support the inference of screw-dislocation in chiral IPNCs.

1. From the view on origins of chirality in inorganic nanocrystals. The chirality in inorganic nanostructures may be induced by five issues listed below, which has been well concluded in many review papers (e.g. Chem. Rev. 2017, 117, 12, 8041-8093 <https://pubs.acs.org/doi/10.1021/acs.chemrev.6b00755>).

- f) The inorganic nanocrystal has chiral shape (in chiral space group)
- g) Interaction with chiral surface ligands
- h) Assembles in chiral patterns
- i) Chiral defects in inorganic nanocrystal
- j) Polarization effects in the inorganic nanocrystal

For the IPNCs in this work, the cubic phase was determined by both HRTEM and XRD patterns, which is consistent with the CsPbBr₃ nanoplates in Alivisatos' work (J. Am. Chem. Soc. 2015, 137, 51, 16008-16011, <https://pubs.acs.org/doi/abs/10.1021/jacs.5b11199>). Even though the tetragonal or monoclinic phase is mixed, they still remain in achiral space group. So, the perfect CsPbBr₃ nanocrystals do not have chiral shape. Then, no chiral ligands were used during the whole synthesis process. There is no ligand-induced chirality here. For (e), these chiral effects often exist with surface plasmon in metal nanomaterials and is not suitable as an explanation here. By exclusion, the chiral defects and chiral assemblies are the most reasonable explanation in this work.

2. Evidence from HRTEM. According to the paper suggested by the reviewer, the screw-dislocation in GaN crystal can be observed under the atomic resolution TEM images in Nat Commun 6, 7266 (2015), <https://www.nature.com/articles/ncomms8266>. These bulk GaN crystals are prepared by expensive epitaxial growth, which benefits a lot for TEM characterization. Additionally, these images are post-processed by average tens

of TEM images with different focus depths. We are also eager for more clear evidence under the micrograph. The atomic resolution for solution-based syntheses CsPbBr₃ nanoplates is very difficult to obtain, as there is only 1-2nm in thickness. Compare to some recent work (e.g. Andry Rogach et al. Adv. Sci. 2019, 6, 1900462. <https://doi.org/10.1002/advs.201900462>; Peidong Yang et al. J. Am. Chem. Soc. 2019, 141, 33, 13028-13032, <https://doi.org/10.1021/jacs.9b06889>), the resolution of TEM images of the inorganic perovskite nanoplate almost in the same level (not reach the atomic resolution). Now, the problem comes into whether we can predict the exists of screw-dislocation from this kind of “bent” crystalline in TEM images. The screw-dislocation induced intrinsic chirality in semiconductor nanocrystals has been systematically studied by Yurri Gunko’s group, and the theoretical explanation (e.g., Nano Lett. 2015, 15, 3, 1710-1715, <https://doi.org/10.1021/nl504369x>) and experimental evidence (e.g., Nano Lett. 2015, 15, 5, 2844-2851, <https://doi.org/10.1021/nl504439w>) can be found in their highly cited papers. For example, the highlighted red points in Figure 5 (A, B) in Nano Lett. 2015, 15, 5, 2844-2851, <https://doi.org/10.1021/nl504439w> is provided for evidence of screw-dislocation in CdSe/ZnS nanoparticles. This revised manuscript, more detailed inference has been updated including the Burgers vector and crystalline plane information. This may help readers to have a more intuitive understanding.

3. The CD strength of IPNCs is about one-tenth of the strength from traditional organic-inorganic hybrid chiral perovskites (e.g. Materials Horizons 2017, 4(5): 851-856, <https://doi.org/10.1039/C7MH00197E>; Nano Letters 2018, 18(9): 5411-5417, <https://doi.org/10.1021/acs.nanolett.8b01616>). Considering the TEM images, only one in the 20-30 nanocrystals was found with dislocation. Such low intensity may belong to the low distribution density of screw-dislocation. Further calculation on CD strength is revised and matches well with the experimental results.
4. The temperature-dependent CD variation via laser treatment. We applied femtosecond

pulse laser (800 nm, 1000hz, 150 fs pulse width) to introduce thermal vibration in IPNCs, and we found the chirality inversion events. This phenomenon excluded the possibility of ligand-induced chirality. Firstly, 800nm (~1.5eV) photon energy cannot be absorbed by organic surfactants and no corresponding reports that show the CD signal from organic molecules got inversed after NIR light irradiation. Additionally, DFT is widely utilized for simulating the chirality induced by ligand-attachment, where the electrons are excited from the nanoparticles to the chiral ligands (ACS Nano 2016, 10, 3, 3809-3815 <https://doi.org/10.1021/acsnano.6b00567>, ACS Nano 2018, 12, 6, 5341-5350 <https://doi.org/10.1021/acsnano.8b00112>). The temperature cannot reverse this mechanism since the temperature changes not the electronic structure of the system, but only the density of states of the energy levels (Figure R1a, details of calculation is added in Supporting Information Section S11). However, the reverse was observed with temperature dependence. The energy diffusion channel should be within the semiconductor nanocrystal. The chirality is thus believed to be introduced by the screw-dislocation of the nanoplates, in a rate of appearance approximated to be one in every twenty. The analytical calculation based on the screw dislocation and the thermal fluctuation to solve a Schrodinger equation for the CD calculation. The chirality will reverse and decay according to the temperature (shown in revised Figure R1b) since the different dislocation states are believed to have different free energy and energy barriers amongst them, which will be reached upon some critical temperature. As for the decay, it will be introduced from the newly introduced thermal fluctuation term, whose physical meaning is the stimulated phonon vibration and is similar to Einstein's phonon model.

Figure R1. (a) Calculation of temperature-dependent CD induced by surface chiral ligands. (b) Numerical data of temperature-dependent CD induced by screw-dislocation with the parameters observed from experiments.

The reviewer suggested the “twin” structure according to the TEM images in (Nat Com. 8, 14547, 2017, doi:10.1038/ncomms14547, <https://www.nature.com/articles/ncomms14547>). However, this work doesn’t provide HRTEM with comparable crystal lattices. Combining the above evidence, we finally provide an explanation of the origin of chirality in IPNCs that there exists screw-dislocation and chiral assembles. That is the best we can do, and we believe that there still some mystery in chiral IPNCs; we hope this work can provide a reference for readers in this research field.

Furthermore, as is always the issue with TEM studies, the areas shown are very small and from the data at hand, it is difficult to evaluate if they are representative. For example, Figure 3 (b) and 4 (c,d) show a single group of chirally assembled nanocrystals each having a different chirality, but it is not possible to see other groups and assess whether they have the same or a different chirality. If the orientations are randomly distributed with a small majority in one direction or another, showing crystal assemblies with different orientations as evidence of orientation inversion is not evidence for a microscopic origin and can be misleading. Similarly, the lines drawn in Figure 4 (e,f) do not follow the same directions. In Figure 4 (e), the red line is parallel to the planes in the crystal, whereas in (f) it is parallel to the crystal in the top and perpendicular to the crystal in the bottom. If the line in (f) were perpendicular to

the crystal planes in the bottom crystal, it would have the same curvature as the line in (e), and no inversion would be observed. This should be addressed.

Reply: Thanks for this comment. As the reviewer claimed, in nanoscience research, sometimes it's not convincing to give judgment from the small areas under microscopic images. In this work, the CD spectrum is the key issue and also the only criterion for learning algorithms in MAOSIC. A microscope like TEM is utilized as a supplementary method for further theoretical analysis. For IPNCs samples synthesized under low temperature (<130°C) condition, it's difficult to give the evidence of screw-dislocation with HRTEM due to the poor crystal quality. So, the chiral assembly patterns can be another authentic evidence for the origin of chirality in IPNCs. It's possible that both L and D oriented IPNCs patterns assembled on one TEM grid, but the detected CD signal may come from the overall result of this sample. The red line in Figure 4f has been revised to be parallel to the crystalline.

Similarly, in Figure 4 (e,f), only one in the 20-30 nanocrystals seen in the lower magnification micrographs show the supposed screw "attachment". If this ratio is representative, is it enough to induce the changes in the CD? It is not obvious to me from the theory section how the CD depends on the number or ratio of chiral nanocrystals, but the strength of the CD seen in Figure 4 (b) is much larger than that predicted by the theory presented in Figure 4 (i), why is this? This should be commented on as well.

Reply: As the reply in the last comment, The CD strength of IPNCs is about one-tenth of the strength from traditional organic-inorganic hybrid chiral perovskites (e.g. Materials Horizons 2017, 4(5): 851-856, <https://doi.org/10.1039/C7MH00197E>; Nano Letters 2018, 18(9): 5411-5417, <https://doi.org/10.1021/acs.nanolett.8b01616>). Considering the TEM images, only one in the 20-30 nanocrystals was found with dislocation. Such low intensity may belong to the low distribution density of screw-dislocation. Further calculation on CD strength is added and matches well with the experimental results. The model in this work describes only one cylindrical nanoplate in unit volume, while in reality, there are, as you mentioned, a

collection of nanoplates and only part of which have screw dislocations. These dislocation-free nanoplates will dilute the circular dichroism of the dislocation-equipped samples. This dilution is assumed to be proportional to the dislocation ratio. For example, the numerical CD strength for NCs sample was 20 mdeg. This value was calculated with certain nanoplates concentration in unit solution volume and all nanoplates attached with screw-dislocation. A constant $k(0 < k < 1)$ is set to describe the ratio of the dislocation-equipped samples among all of them. The observed circular dichroism will be the product of k of the result of our calculation. Constant k was determined according the TEM image. As described above, k should be set in range of (1/20, 1/30) to match the experimental results. Above discussion has been added in “Temperature dependence of the chiral intensity” section of the revised manuscript to avoid misunderstanding.

A micrograph with a magnification somewhere between that of Figure 4 (e) and (g) could be included, and the nanocrystals with the additional growth could be highlighted along with their orientation to give a better understanding of the differences in orientation.

Reply: The micrograph with a magnification between that of Figure 4(e) and (g) can be found in Figure S16. According to Figure 4h, the additional growth of IPNCs (dark crystals in large size) can both be parallel or perpendicular to the Z-axis of original nanoplates around them.

Similar to the initial version of the paper, it is difficult to see how the only change in microstructure after irradiation is the presence of a small amount of Cs₄PbBr₆ near the edges of the CsPbBr₃ nanocrystals (Figure 4 (g,h)). After irradiation, particles or rods of several hundred nm length are formed, and the relative ratios of the XRD peaks change significantly. It is clear that the overall intensity drops and a secondary peak appears near 30 degrees, with almost the same intensity as the original peak at that angle. The peak near 15 degrees also shifts to lower angles and there are additional significant peaks near 20, 26, 28, 33, and 37 degrees comparable in intensity to the one near 15 degrees. The NEB calculations shown in SI S7 do not obviously support the change in the XRD pattern and microstructure observed in Figure 4 (g,h). All of this should be addressed.

Reply: The two peaks around 30 degrees after irradiation agrees well with the XRD pattern of the orthorhombic CsPbBr₃ phase (Chemistry of Materials 2019, 31(17): 6642-6649). The key evidence of the Cs₄PbBr₆ phase is the (012) peak at 12 degrees in the XRD pattern. The additional peaks near 20, 26, 28, 33 degrees can all be checked in the reference pattern number 73-2478 for Cs₄PbBr₆ crystals. The phase transition and regrowth from CsPbBr₃ to CsPbBr₃/Cs₄PbBr₆ core/shell nanostructure in ion-rich solution under heating was reported in many works. (e.g. J. Phys. Chem. Lett. 2019, 10, 18, 5302-5311 <https://doi.org/10.1021/acs.jpcclett.9b01552>. Cryst. Growth Des. 2018, 18, 10, 6133-6141. <https://doi.org/10.1021/acs.cgd.8b01013>). The temperature-driven transformation of CsPbBr₃ nanoplates to Cs₄PbBr₆ nano-assembly was investigated in recent work (*Nano Lett.* 2020, XXXX, XXX, XXX-XXX <https://doi.org/10.1021/acs.nanolett.9b05036>). This work further confirmed the reproducibility of our experiments and indicating that the reaction with an excess of amines in solution should be the main mechanism. The above discussion has been added in the revised manuscript. The NEB calculation here in Supporting Information Section S7 provides a feasible path from CsPbBr₃ to Cs₄PbBr₆ at the edge of CsPbBr₃ nanocrystal.

When reading through the ‘Eigenfunction and eigenvalues’ section, I find it very difficult to follow which parts of it are general, already published knowledge, and which parts are new derivations used to explain the current system. The section is called ‘Eigenfunction and eigenvalues’ but these expressions are not used anywhere in the text other than in the title of the section. The whole derivation has very few explanations. For example, it is not clear to me from the text why rho is an important parameter or why the Whittaker-M function blows up for the negative mu when rho = 0, but not for the positive mu. Similarly, the Cotton effect is introduced without any explanation of what it is or what it means for the properties of the materials. It would be good to specify which parts of the section are new derivations, and which are based on literature, and include citations at the appropriate places, for example when introducing the Whittaker-M function, as well as other places.

Reply: Thanks for this comment. We added the statement of the new derivations in the revised manuscript, which has been highlighted and shown below.

To describe the thermal fluctuation induced by temperature, a harmonic potential is added to the Schrodinger equation to modify the one obtained earlier by other researchers through pure dislocation analysis (Nano letters 2015, 15(3): 1710-1715; Annalen der Physik 1999, **8**(3): 181-189). The harmonic potential here refers to the thermal oscillation, which can be denoted the same as Einstein's model $\hbar\omega$ for the phonon energy, and is proportional to the "temperature". The magnitude of the energy term should thus be $n\hbar\omega$, where n is the density of states of the phonons. According to the textbook written by Mr. Charles Kittel (Introduction to solid-state physics. Fifth edition. United Kingdom.), this density of states is the Bose-Einstein distribution $(e^{\hbar\omega/k_B T} - 1)^{-1}$. When $k_B T \gg \hbar\omega$, the formula $n\hbar\omega$ will reduce to $k_B T$. This is classic as the Einstein model for the heat capacity approaches the Dulong Petit Model at high temperatures.

This temperature, though, is not the traditional temperature of the surroundings. Since the inversion of the chirality of the nanoplates was observed after the excitation of laser, the heat was absorbed only locally by the nanoplates. Thus, the energy converts only to its phonon oscillation.

The Cotton effect is the characteristic change in optical rotatory dispersion and/or circular dichroism in the vicinity of an absorption band of a substance. In a wavelength region where the light is absorbed, the absolute magnitude of the optical rotation at first varies rapidly with wavelength, crosses zero at absorption maxima, and then again varies rapidly with wavelength but in the opposite direction. It is positive if the optical rotation first increases as the wavelength decreases while negative if the rotation first decreases. As the Cotton Effect is not the key issue in this work, this part is removed from the revised manuscript.

As for the mathematical properties of the Whittaker function, it is a special function whose

definition can be found in the book written by Mr. E.T. Whittaker and G. N. Watson (*A Course of Modern Analysis* (Whittaker, E. T., & Watson, G. N. (1996). A course of modern analysis. Cambridge university press.)) Detail information about the Whittaker function is in *Chapter XVI, The Confluent Hypergeometric Function*. (It is the reprinted fourth version of the book that we were reading.) One can refer to the asymptotic expansion (NIST Digital Library of Mathematical Functions. <http://dlmf.nist.gov/>, Release 1.0.25 of 2019-12-15. F. W. J. Olver, A. B. Olde Daalhuis, D. W. Lozier, B. I. Schneider, R. F. Boisvert, C. W. Clark, B. R. Miller, B. V. Saunders, H. S. Cohl, and M. A. McClain, eds.) of it around zero:

$$M_{\kappa,-\mu}(\rho) \rightarrow \rho^{-\mu+\frac{1}{2}}(1 + O(\rho)).$$

Since $\mu > 1/2$, this thing would blow up to infinity when $\rho \rightarrow 0$. The citation will be added in the main text. Thanks for your comment.

The explanations in the Temperature Dependence section are better, but the concept of omega seems to appear just before Equation 9 and is not really explained. From the text, it is not clear to me how the Burger's vector, the vector that describes the screw dislocation, changes with temperature. Does the screw dislocation oscillate back and forth with changing temperature, or from the influence of an external electromagnetic field? It is claimed that the CD converges, but the graph in Figure 4 (i) stops at 250 C while having a negative slope and before any convergence is shown.

Reply: The Burger's vector measures the magnitude of the screw dislocation, and thus in our model, it measures the intensity of the circular dichroism. There was no external electromagnetic field, but, according to the assumptions made in the main text, there are several dislocation states of the nanoplates, just like the cubic and orthorhombic lattice of the perovskite. The free energy of them are different, but only upon reaching the energy barrier between them, will the nanoplates be able to transit from its current state to another one with lower free energy, if needed. Thus, with the temperature increasing, the CD signal will oscillate back and forth. This agrees with the experiments since the reversal of CD the signal was observed; this also agrees with the model since, if the sign of β is reversed, the Cotton

effect observed will also reverse. As for the magnitude of the data and the convergence, sorry that the experimental data was not used in the early version of our calculation. The calculation was done again with the experimental parameters. From the TEM images (Figure 3c in the main text), $|\mathbf{b}| = 0.4\text{nm}$, $R = 2\text{nm}$, $L = 5\text{nm}$. Since the penetration depth of the circularly polarized light was measured to be $\sim 1\text{nm}$, and the absorption peak is around $\omega \sim 10^{15}\text{Hz}$, the imaginary part of the refractive index will reach ~ 100 . This follows from the attenuation constant from the Beer-Lambert Law $I = I_0 e^{-\alpha x}$. The penetration depth is defined as $\delta_p = 1/\alpha$, and $\alpha = \frac{\Im(n_{\text{complex}})\omega}{c}$. Adapting these parameters into the calculation of $\Gamma(\omega)$ shown in supporting information S10, with the assumption that the excitation only takes place from the ground state to the first excited state, i.e. from $(n, l, k) = (1, 0, 0)$ to $(n, l, k) = (1, \pm 1, k)$, and taking the absorption peak ω to be the same as the one at 273K in the entire temperature range (assuming no change to the absorption peak during this process, agrees with the experiments), one obtains the figure as follows.

Revised Figure 4i. Numerical data with the parameters observed from experiments.

In SI S8 it is not clear what the green and blue lines represent in Figure S15 (a,c), and the scale bars in Figure S16 are too small to read. The 'w' in milli-watt should also be capitalised as 'mW'.

Reply: Thanks for your comment. The legend in Figure S15 has been added. Scale bar in

Figure S16 is adjusted to a visible scale. “w” has been correlated to “mW”

I will not go into details regarding the language editing, but I highly recommend thorough editing for language and formatting.

Reviewer #3 (Remarks to the Author):

I attached a PDF file where I highlighted grammar/spelling mistakes that should be reviewed and fixed before submission. Also, in Figure S6b there is a part labelled as “Y-shaped value” should this be “Y-shaped valve”?

Reply: Thanks for helping us checking the grammar/spelling mistakes in the manuscript. The “Y-shape value” should be “Y-shaped-valve”.

One confusion I have is about the two types of chirality discussed on pages 3, 9 and in Figure 4. The authors refer to “chiral assembled structures” but my understanding was that the chirality measurements are done in solution.

1. Is there any evidence that the assembled structures observed on the TEM grid correlate to an assembly in solution? When NCs are deposited on TEM grids they can assemble into superlattices, which are not correlated to an assembly in solution, but rather to the deposition technique.

Reply: The assembled structures here refer to nanostructures with a certain number of nanoparticles, for example, dimers, tetramers, and nematic shapes. (Mater. Chem. Front., 2018, 2, 662—678) With fine control over the synthesis, these chiral nanostructures can be obtained in solution. Well, admittedly, like what the reviewer concerns in comment 1, in the TEM measurements, sometimes the deposition technique may bring in some artifact for “assembly”, but considering the great number of publications of these nanostructures and extensive body of review papers concerning on the synthesis, origin of chirality of these

materials, we do believe such “chiral assembled structures” in solution are real and reproducible. (Please see Chem. Rev. 2017, 117, 8041–8093; Adv. Mater. 2019, 1905975).

2. On page 9, the authors refer to a “high concentration”. The concentration controlled in the flow reactor is that of the precursors, but here they seem to be talking about the final concentration of NCs in solution. This point should be cleared up. Also, if the chirality depends on the concentration of product (As suggested in the 2nd to last sentence on page 9) then why can't you adjust the concentration of NCs in solution after synthesis to increase the chiral signature detected? I think this is a different point to how the precursor concentration influences the reaction path and therefore chirality.

Reply: The concentration refers to precursors concentration, which defined as C in the manuscript. The final concentration of NCs in solution was affected by the initial synthesis condition including C and reaction temperature (T). In general, with C increases, the concentration of NCs in solution increases, and some other properties also vary with it, including the average size (Lab Chip, 2007, 7, 1434-1441 <https://doi.org/10.1039/B711412E>), size distribution (CrystEngComm, 2017, 19, 6694-6702 <https://doi.org/10.1039/C7CE01406F>) and quantum efficiency (J. Phys. Chem. Lett. 2015, 6, 21, 4360-4364 <https://doi.org/10.1021/acs.jpcllett.5b02011>). In Figure 2, the CD intensity increases with the C increases, which is due to the increase of NCs concentration. But when $C > 1 \text{ mmol/ml}$, CD intensity decreases, the too high concentration may be harmful for the synthesis reaction (J. Am. Chem. Soc. 2002, 124, 9, 2049-2055 <https://doi.org/10.1021/ja017002j>). So, the CD intensity can increase with the increase of NCs concentration in solution, but from the matter of fact, of NCs concentration cannot increase unlimitedly with C increases.

3. In Figure 4c and d: Was this sample irradiated on the TEM grid? If not, do you have reason to believe that the orientation of the nanorods on the grid is more influenced by the previous irradiation in solution as opposed to the dynamics of solvent drying during deposition?

Reply: Thanks for the comment. The sample is irradiated in solution. First, technically, it is not suggested to put TEM grid under a pump laser because the strong laser will destroy the grid. Last but not least, it has been verified that light radiation on a solution of nanocrystals can have access to form chiral crystals which expresses optical chirality. (Nature Materials, 2014, 14(1):66-72; Sci. Adv. 2017;3: e1601159.)

On page 8, “the nanostructure of nanoplates kept unchanged during the laser treatment”. I consider the screw dislocation to be part of the nanostructure of the NCs, so saying it is unchanged even though you are inverting the chirality of the screw dislocation is misleading.

Reply: Thanks for your comment. The description “nanoplates kept unchanged” is not clear. It has been changed as: The peak wavelength and intensity of optical absorption vary less than 1% (the insert pictures in both Figures 4a and b), and the CD strength after irradiation was similar to the original intensity. Combining with the TEM images (Figure 4 c-f), the shape and size of nanoplates kept unchanged during the laser treatment.

For the temperature dependence of the chirality, is there a reason why the 800nm laser was chosen to induce the chirality change instead of just heating the sample to various temperatures? Based on your model in 4i, the temperature chosen should give you the control to form D or L based on the temperature, instead of just being able to switch from one to the other. If you have a mixed sample (containing both D and L), do all the NCs invert? Or do you see that they all tend to the same D or L chirality after laser exposure?

Reply: We chose the 800nm femtosecond pulse laser to enhance the thermal vibration of the dislocations in nanocrystals. The femtosecond laser source is widely applied in fast events investigation of like electron dynamics, and in some laser-assisted surgery with high precise and tiny trauma requirements. Similarly, we choose it to affect the crystal dynamics because it may cause less damage to the sample itself. 300nm laser

was also applied to treat the sample. Irreversible degradation and phase transition were found, agree with related reports (Li J, Bo B, Gao X. The degradation of structure and luminescence in CsPbBr₃ perovskite nanocrystals under UV light illumination. *AIP Conference Proceedings* 2018, **2036**(1): 030018.). No chirality inversion was observed and CD signal reduced significantly.

The principle of direct heating is to heat the solvent and then energy transferred to NCs in solution. As the synthesized sample was in-situ tested without deep purification, the ligand-rich solution direct heating broke the sample (*Nano Lett.* 2020, XXXX, xxx, xxx-xxx <https://doi.org/10.1021/acs.nanolett.9b05036>) and CD intensity reduced without inversion. Additionally, the dislocation formation belongs to the “non-equilibrium” process (Müller G, Métois JJ, Rudolph P. *Crystal Growth - From Fundamentals to Technology*. Elsevier Science, 2004), in which cases the light induction takes more advantage than the direct heating does (Nagaev E. Light-induced phase transitions. *Zh Eksp Teor Fiz* 1981, **80**: 2346-2355).

The CD inversion direction is related to the chirality of origin samples. From the experimental results, both of the inversion from L to D and from D to L can be achieved. Here we just plot one situation in Fig 4i, which is D to L. So of course, if the original sample at room temperature was L, the temperature dependence should change into L to D. And from our theoretical work, we find this CD inversion is the straightforward result from the screw-dislocation induced mechanism (comparing the calculation results in manuscript section “Temperature dependence of the chiral intensity” and Supporting Information Section S11).

In Figure 4 caption it says that the chirality changes twice, but it looks like it inverts 3 times (L→D, D→L, and then L→D again) in 4i. Based on this model do believe the chirality is determined by the synthesis temperature?

Reply: Thanks for your comment. Figure 4i has been revised. Only one inversion is obtained

according to the revised calculation, which matches well with the experimental data. The temperature here is not the synthesis temperature. It is a calculated temperature which indicates the energy obtained from the 800nm laser. As mentioned in the early session of the reply, the phonon energy adopted the Einstein model, which approaches the classic Dulong Petit model. The phonon energy is then $n\hbar\omega = (e^{\hbar\omega/k_B T})^{-1} \hbar\omega \rightarrow k_B T$ when $\hbar\omega \ll k_B T$. During the phonon vibration, the displacement of the atoms is only several angstroms and is denoted in the Schrodinger equation as $\propto r^2$. By equating these two expressions of the phonon energy, i.e. the phonon vibration via the harmonic potential calculation and the energy approached to the temperature, one is able to get a fictional temperature to describe the intensity of the thermal-induced phonon behavior in the nanoplates.

REVIEWERS' COMMENTS:

Reviewer#2

Thank you to the authors for their comprehensive revision of the manuscript titled 'Autonomous Discovery of Optically-Active Chiral Inorganic Perovskite Nanocrystals Through Intelligent Cloud Lab'. It is clear that significant effort has gone into addressing the reviewer comments and the manuscript has improved significantly in terms of precision, clarity, and readability. All in all, the authors have referenced a significant amount of literature that support their hypothesis of screw dislocations while at the same time reducing the strength of the claims to suit those made in the literature. In particular, the focus has been shifted from the presence of screw dislocations to the presence of circular dichroism, with the presence of screw dislocations being a supporting, and slightly tentative but supported by literature, explanation.

The authors have specified the relative presence of the nanoparticles with the supposed screw dislocations and correlated this to the strength of the circular dichroism compared to their theoretically determined values, which seems to fit.

Similarly, the theory section is much easier to follow and the explanations detailing the consequences of the theory have been included, meaning that it is also useful for the general audience.

Reading through the manuscript, there are a few sections that could be clarified or amended slightly, and there is still an overall need for a thorough editing for language, as some sentences or paragraphs are either ambiguous or difficult to understand.

Overall, I think the authors have a very interesting approach and this is an interesting piece of research that highlights a relatively un-studied property of inorganic perovskite nanoparticles, and I would recommend publication after addressing the comments and an editing for language clarity.

The lines below refer to the line number in the latest pdf version. Some of these comments refer to language and spelling, but it is not exhaustive.

61: MAOSIC does not seem to be defined in the main text after the abstract, which it probably should be.

137: 'Algorisms' should probably be 'algorithms'

146: Should 'pool' be 'poor'?

177: There seems to be an 's' inserted in the sentence.

197: Where is the HOMO/LUMO plot? Is it in a reference? If so, a better wording than referring to a plot in a reference would probably be appropriate. If it is meant to be included in the main text, I can't find it.

210: Was the laser applied while the nanoparticles were in solution? I cannot find the description of where the laser was applied.

Figure 4 (i): What wavelength was used for this figure? Is it the temperature derived from the theory as not the environmental temperature, but the temperature of the particles?

314-322: This is a very long sentence and should probably be broken into several smaller sentences. The sentence seems to mention horizontal and vertical axes but there is not plot associated with this paragraph or Supporting Section S10. Is there meant to be a plot, or what do the axes refer to?

350: The conclusion still fairly strongly states that screw dislocations are the cause of the chirality while the text has been updated to specify that it is probably likely due to chirality induced by defects, screw dislocations being one possibility. The conclusion should reflect this.

Supplementary information:

Figure S1: Should the 'COMMAND' and 'INFO' arrows be pointing the opposite directions? The command going to the control interface, and the info coming out of it. Similarly, the first arrow pointing right spells 'COMMOND', which should probably be 'COMMAND'.

185: The whole paragraph in S2 is difficult to understand. This should probably be edited for language.

426-432: This paragraph discusses a plot but I cannot find the plot in the text. It should probably

be included.

Figure S15: What are the OD and A on the vertical axes? They are not defined.

Dear reviewer

Thank you for your kind comments concerning our manuscript entitled “Autonomous Discovery of Optically Active Chiral Inorganic Perovskite Nanocrystals Through an Intelligent Cloud Lab” (Manuscript ID: **NCOMMS-19-33249D**). Those comments are all valuable and very helpful for revising and improving our paper, as well as the essential guiding significance to our research. We have studied comments carefully and have made a correction, which we hope to meet with approval. We also applied the language editing service of *Springer Nature Editing* to improve the quality of the manuscript. The revised portion is highlighted in yellow in the manuscript and supporting information.

REVIEWERS' COMMENTS:

Reviewer#2

Thank you to the authors for their comprehensive revision of the manuscript titled ‘Autonomous Discovery of Optically-Active Chiral Inorganic Perovskite Nanocrystals Through Intelligent Cloud Lab’. It is clear that significant effort has gone into addressing the reviewer comments and the manuscript has improved significantly in terms of precision, clarity, and readability. All in all, the authors have referenced a significant amount of literature that support their hypothesis of screw dislocations while at the same time reducing the strength of the claims to suit those made in the literature. In particular, the focus has been shifted from the presence of screw dislocations to the presence of circular dichroism, with the presence of screw dislocations being a supporting, and slightly tentative but supported by literature, explanation.

The authors have specified the relative presence of the nanoparticles with the supposed screw dislocations and correlated this to the strength of the circular dichroism compared to their theoretically determined values, which seems to fit.

Similarly, the theory section is much easier to follow and the explanations detailing the consequences of the theory have been included, meaning that it is also useful for the general audience.

Reading through the manuscript, there are a few sections that could be clarified or amended slightly, and there is still an overall need for a thorough editing for language, as some sentences or paragraphs are either ambiguous or difficult to understand.

Overall, I think the authors have a very interesting approach and this is an interesting piece of research that highlights a relatively un-studied property of inorganic perovskite nanoparticles, and I would recommend publication after addressing the comments and an editing for language clarity.

The lines below refer to the line number in the latest pdf version. Some of these comments refer to language and spelling, but it is not exhaustive.

61: MAOSIC does not seem to be defined in the main text after the abstract, which it probably should be.

Reply: Thanks for reminding. Definition MAOSIC has been added in main text: Driven by the customized materials acceleration operating system in cloud (MAOSIC) platform, this cloud lab has provided an authentic synthetic platform for researchers across the world.

137: 'Algorisms' should probably be 'algorithms'

Rely: Correction has been made: Guided by algorithms, the light yellow points were sparsely distributed on the 2-D map to divide the parameter space into similar boxes, which indicates the high randomness of sampling at the initial state of the reaction search.

146: Should 'pool' be 'poor'?

Rely: Correction has been made: However, the poor crystallization and dispersion state limited the sample quality and CD strength.

177: There seems to be an 's' inserted in the sentence.

Rely: Correction has been made: The X-ray diffraction (XRD) patterns in Figure 3a demonstrate a significant component of the cubic CsPbBr₃ perovskite phase (JCPDS card No. 54-0752).

197: Where is the HOMO/LUMO plot? Is it in a reference? If so, a better wording than referring to a plot in a reference would probably be appropriate. If it is meant to be included in the main text, I can't find it.

Reply: The distribution of HOMO/LUMO is according to reference #38 (Nature 2018, 561(7721):

88-93. The description has been improved to avoid misunderstanding: Referring to the orbital distribution in the CsPbBr₃ nanocluster³⁸, the locations of the highest occupied molecular orbital (HOMO) and lowest unoccupied molecular orbital (LUMO) around Pb and Br atoms indicate that the screw dislocation among Pb-Br crystals may result in significant optical activity. By this inference, the origins of the CD signal come from the screw orbital geometry-induced electron scattering, as discussed in many studies^{39, 40, 41}.

210) Was the laser applied while the nanoparticles were in solution? I cannot find the description of where the laser was applied.

Reply: The sample was treated by laser in solution. The description has been updated: As shown in Figure 4a, after 300 seconds of irradiation in solution, the CD signal of the local optimal sample near the first exciton absorption peak showed inversion from L to D.

Figure 4 (i): What wavelength was used for this figure? Is it the temperature derived from the theory as not the environmental temperature, but the temperature of the particles?

Reply: The wavelength of femtosecond laser is 800nm. The energy of absorption in CD calculation is set as 2.75eV (around 450nm). As you said, the temperature used for calculation is the temperature of the IPNCs, not the surrounding solution.

314-322: This is a very long sentence and should probably be broken into several smaller sentences. The sentence seems to mention horizontal and vertical axes but there is not plot associated with this paragraph or Supporting Section S10. Is there meant to be a plot, or what do the axes refer to?

Reply: Thanks for reminding. The horizontal and vertical axes should refer to the axes in Figure 4i, correlation has been made. This long sentence has been rewritten into: Adapting these parameters into the calculation of $\Gamma(\omega)$ in Supporting Information S10, the temperature dependence of the CD intensity can be calculated. Assumptions made for the calculations are that the excitation takes place only from the ground state to the first excited state, i.e., from $(n, l, k) = (1, 0, 0)$ to $(n, l, k) = (1, +1, k)$, and the absorption peak ω is the same as that at 273 K over the entire temperature range (assuming barely no change to the absorption peak during this process, which agrees with the experiments). The resulting plot is shown in Figure 4(i), where the horizontal axis is the equivalent

temperature adopted from Einstein's model by equating $k_B T$ to the energy of the harmonic oscillators, which is proportional to the square of the displacement of the lattice, which is $\sim 1 \text{ \AA}$, and the vertical axis is the exact $CD(mdeg)$ value.

350: The conclusion still fairly strongly states that screw dislocations are the cause of the chirality while the text has been updated to specify that it is probably likely due to chirality induced by defects, screw dislocations being one possibility. The conclusion should reflect this.

Reply: Thanks for your suggestion. New conclusion has been updated by adding: Screw dislocation and chiral assemblies were captured by TEM to provide a possible explanation of the origin of chirality.

Supplementary information:

Figure S1: Should the 'COMMAND' and 'INFO' arrows be pointing the opposite directions? The command going to the control interface, and the info coming out of it. Similarly, the first arrow pointing right spells 'COMMOND', which should probably be 'COMMAND'.

Reply: The spelling error "COMMOND" has been corrected ad "COMMAND". The arrow direction has been checked and updated in Figure S1.

Revised Figure S1. Scheme of MAOSIC. Five modules were designed and intercalated with each other for autonomous materials discovery.

185: The whole paragraph in S2 is difficult to understand. This should probably be edited for language.

Reply: The language of S2 has been improved: Here, we tested the performance of different networks applying for the cloud lab (Shown in Figure S3 and Table S3). Under 5G network (provided by China Unicom & Ericsson), 405Mbps data rate and 19ms Ping significantly enhanced the preciseness of remote control of experimental platform.

426-432: This paragraph discusses a plot but I cannot find the plot in the text. It should probably be included.

Reply: The description has been corrected and no plot is provided here. By comparing the magnitude of $E_{thermal}$ and $E_{dis} = e_{stress}l$ per site, it could be observed that: as the radius growing, i.e. the morphology approaches 2-D, the energy is at first very small because of the size, the energy of dislocation then increases but falls below the temperature effect afterwards, which means the defected morphology is more likely to occur at if the morphology is more similar to 2-D; as for the temperature, the energy of dislocation also falls below the thermal energy after reaching some threshold, indicating that the defects are more likely to occur at higher temperature.

Figure S15: What are the OD and A on the vertical axes? They are not defined.

Reply: The definition of ΔOD and ΔA has been added in figure caption: Transient absorption spectrum (TAS) at 5ps (ΔOD means the difference of optical density before and after the optical excitation of sample) and fitting results of average carries lifetime (ΔA means the difference of absorption) for global optimal sample (a) (b) after 800nm laser and (c) (d) 300nm laser irradiated. All characterization was taken under 800nm pump laser with 8mW power.

We tried our best to improve the manuscript and made some changes in the manuscript. These changes will not influence the content and framework of the paper. And detailed changes are highlighted in yellow in revised paper.

We appreciate for reviewers' warm work earnestly, and hope that the correction will meet with approval.

Once again, thank you very much for your comments and suggestions.

Xi Zhu

Deputy Director

Center for AI General Application

Shenzhen Institute of Artificial Intelligence and Robotics for Society (AIRS),

The Chinese University of Hong Kong, Shenzhen, Guangdong, P.R. China. 518172

zhuxi@cuhk.edu.cn